# MicroRNA-146a limits tumorigenic inflammation in colorectal cancer

Lucien P. Garo [1,2], Amrendra K. Ajay [3,4], Mai Fujiwara [1,4], Galina Gabriely[1], Radhika Raheja[1], Chantal Kuhn[1], Brendan Kenyon[1], Nathaniel Skillin [1], Ryoko Kadowaki-Saga[1], Shrishti Saxena[1] & Gopal Murugaiyan [1✉]

Chronic inflammation can drive tumor development. Here, we have identified microRNA-146a (miR-146a) as a major negative regulator of colonic inflammation and associated tumorigenesis by modulating IL-17 responses. MiR-146a-deficient mice are susceptible to both colitis-associated and sporadic colorectal cancer (CRC), presenting with enhanced tumorigenic IL-17 signaling. Within myeloid cells, miR-146a targets RIPK2, a NOD2 signaling intermediate, to limit myeloid cell-derived IL-17-inducing cytokines and restrict colonic IL-17. Accordingly, myeloid-specific miR-146a deletion promotes CRC. Moreover, within intestinal epithelial cells (IECs), miR-146a targets TRAF6, an IL-17R signaling intermediate, to restrict IEC responsiveness to IL-17. MiR-146a within IECs further suppresses CRC by targeting PTGES2, a PGE2 synthesis enzyme. IEC-specific miR-146a deletion therefore promotes CRC. Importantly, preclinical administration of miR-146a mimic, or small molecule inhibition of the miR-146a targets, TRAF6 and RIPK2, ameliorates colonic inflammation and CRC. MiR-146a overexpression or miR-146a target inhibition represent therapeutic approaches that limit pathways converging on tumorigenic IL-17 signaling in CRC.

[1] Ann Romney Center for Neurologic Diseases, Department of Neurology, Brigham and Women's Hospital and Harvard Medical School, Boston, MA, USA. [2] Boston University School of Medicine, Boston, MA, USA. [3] Renal Division, Department of Medicine, Brigham and Women's Hospital and Harvard Medical School, Boston, MA, USA. [4] These authors contributed equally: Amrendra K. Ajay, Mai Fujiwara. ✉email: mgopal@rics.bwh.harvard.edu

Chronic inflammation plays a key role in the initiation and malignant progression of several cancers, including colorectal cancer (CRC)[1–3]. Interleukin-17 (IL-17) is one of the major inflammatory cytokines recently implicated in both inflammation-associated and sporadic colon cancer models, as well as human CRC[4–11]. In fact, elevated IL-17 has been negatively correlated with CRC patient survival and even linked to resistance to both classical cytotoxic drugs and targeted therapeutics[7,9–12]. Functionally, IL-17 has been shown to exert tumorigenic signaling directly within intestinal epithelial cells (IECs) to promote CRC development and progression[7]. IL-17 receptor (IL-17R) signaling within IECs activates nuclear factor κ-light-chain-enhancer of activated B (NF-κB) cells and mitogen-activated protein kinases (MAPKs), which promote survival pathways required for the growth of premalignant lesions and subsequent tumors[7]. Correspondingly, inactivation of IL-17R and/or NF-κB and MAPKs within IECs can substantially limit CRC[13,14]. However, the molecular regulatory mechanisms within IECs that limit tumorigenic IL-17R signaling to control CRC development remain unclear.

CD4[+] Th17 cells, γδ[+] T cells, and innate lymphoid cells (ILCs) represent the major sources of IL-17 in the tumor microenvironment in the gut[15]. IL-17 in these cell populations is mainly induced by myeloid cells, such as dendritic cells (DCs) and macrophages (MΦs), which respond to microbial products by releasing IL-17-promoting cytokines, for instance IL-1β, IL-6, and IL-23[16]. Among these cytokines, IL-23 is indispensable for the expansion of IL-17 producers in the colon[17]. Recent work has shown that nucleotide-binding oligomerization domain-containing protein 2 (NOD2) signaling in DCs effectively induces IL-17-promoting cytokines, including IL-23, and promotes colonic IL-17[17,18]. Although IL-17 has been linked to tumor progression and therapy resistance in CRC, the molecular regulatory mechanisms that control IL-17 expression in the colonic microenvironment and CRC development remain largely unknown.

MicroRNAs (miRNAs) are a class of small non-coding RNAs that posttranscriptionally regulate gene expression by binding to the 3′-untranslated region (UTR) of target mRNA sequences to prevent translation or promote mRNA degradation[19]. MiRNAs play a key role in many physiological processes, including immune system development and function[20]. Emerging evidence suggests that miRNAs can modulate the magnitude of inflammatory responses, and has associated miRNAs with various inflammatory human diseases, including inflammatory bowel disease (IBD) and cancer[21,22]. Recently, it has been shown that polymorphisms in the miRNA, miR-146a, are associated with susceptibility in CRC[23] and IBD patients[24], and that miR-146a levels are altered in human CRC tissues[25,26]. In fact, CRC patients with high colonic miR-146a have significantly longer total survival times[23,26]. Although miR-146a has been associated with clinical outcomes in CRC, the exact function and therapeutic potential of miR-146a in controlling tumor-promoting inflammation and CRC are not known.

In this study, we have identified a critical regulatory role for miR-146a in preventing destructive colonic inflammation and associated tumorigenesis via modulation of IL-17 responses. Specifically, our data show that miR-146a prevents intestinal inflammation and CRC by two interlinked mechanisms as follows: (1) by limiting myeloid cell-mediated inflammatory IL-17 production and (2) by inhibiting tumorigenic IL-17R signaling in IECs. Mice deficient in miR-146a either globally, specifically within myeloid cells, or specifically within IECs, present with enhanced IL-17 signaling and severe CRC. Importantly, preclinical administration of miR-146a mimic or direct inhibition of miR-146a targets can ameliorate colonic inflammation and CRC.

## Results

### MiR-146a deletion promotes colonic inflammation and cancer.

To study the function of miR-146a in colonic inflammation, we monitored dextran sodium sulfate (DSS)-induced colitis in wild-type (WT) and miR-146a-deficient ($-/-$) mice (Fig. 1a). We found that miR-146a$^{-/-}$ mice lost significantly more body weight with 100% mortality, in contrast to 0% mortality in WT mice (Fig. 1b, c). Accordingly, miR-146a$^{-/-}$ colons were characterized by severe inflammation with complete loss of crypt architecture and high infiltration of inflammatory cells (Fig. 1d), resulting in significantly higher histolopathogical scores (Fig. 1e). In vivo assessment of gut permeability revealed a significant increase in serum fluorescein isothiocyanate (FITC)-dextran in miR-146a$^{-/-}$ mice during DSS-induced colitis, indicating impairment of intestinal barrier function (Fig. 1f). The dramatic effects of miR-146a deficiency on susceptibility to DSS-induced colitis in mice prompted us to investigate the role of miR-146a in inflammation-associated tumorigenesis using a mouse model of colitis-associated CRC[27]. For this, we employed an azoxymethane (AOM) + DSS-driven CRC model (Fig. 1g), in which mice were treated with the carcinogen AOM, followed by three cycles of low-dose (2%) DSS treatment. Although WT mice had few small tumors restricted to the distal colon, miR-146a$^{-/-}$ mice exhibited a significantly increased number of tumors, which were larger and extended farther proximally in the colon (Fig. 1h–j). In contrast to miR-146a$^{-/-}$ mice, WT mice showed only few low-grade adenomas (Fig. 1k). These results demonstrated that miR-146a is critical for protection against colitis and colitis-associated CRC.

Next, we determined the cellular compartments responsible for the phenotype observed in miR-146a$^{-/-}$ mice. For this, we created WT and miR-146a$^{-/-}$ bone marrow (BM) chimeras (Supplementary Fig. 1a) to assess the contribution of miR-146a within immunologic and non-immunologic (including intestinal epithelial) compartments, and confirmed comparable levels of reconstitution in all groups (Supplementary Fig. 1b). We found that miR-146a deficiency in both immune and non-immune compartments appeared to modulate colitis. This was demonstrated by increased weight loss, mortality, and histopathological scores in either WT mice receiving miR-146a$^{-/-}$ BM or miR-146a$^{-/-}$ mice receiving WT BM, as compared to WT mice with WT BM (Fig. 1l–o). Together, these data suggested miR-146a deficiency in both immune cells and non-immune cells confers susceptibility to colonic inflammation and inflammation-associated CRC.

### Myeloid cell-specific deletion of miR-146a promotes CRC.

To identify inflammatory mediators that cause severe colonic inflammation in miR-146a$^{-/-}$ mice, we profiled inflammatory cytokines and chemokines from colitic WT and miR-146a$^{-/-}$ mice. Consistent with increased inflammatory cell infiltration (Fig. 1d), various proinflammatory cytokine and chemokine markers were grossly elevated in miR-146a$^{-/-}$ mice (Fig. 2a). Interestingly, we found a particularly strong enhancement of IL-17, a major cytokine implicated in CRC development[7,9–12]. In addition, we found enhanced levels of IL-17-promoting cytokines (e.g., IL-6, IL-23, and IL-1β), which are known to be secreted by myeloid cells[17,28]. We also observed increased expression of the chemokine, CCL2, and cytokines, CSF1 and CSF2, which promote recruitment of myeloid cells to the site of inflammation[29,30]. Similarly, within CRC tissue from miR-146a$^{-/-}$ mice, we detected upregulation of IL-17 and IL-17-promoting cytokines, as well as the myeloid cell-recruiting chemokine/cytokines CCL2, CSF1, and CSF2 (Fig. 2b). Together, these data implicated that susceptibility to colonic inflammation and inflammation-associated

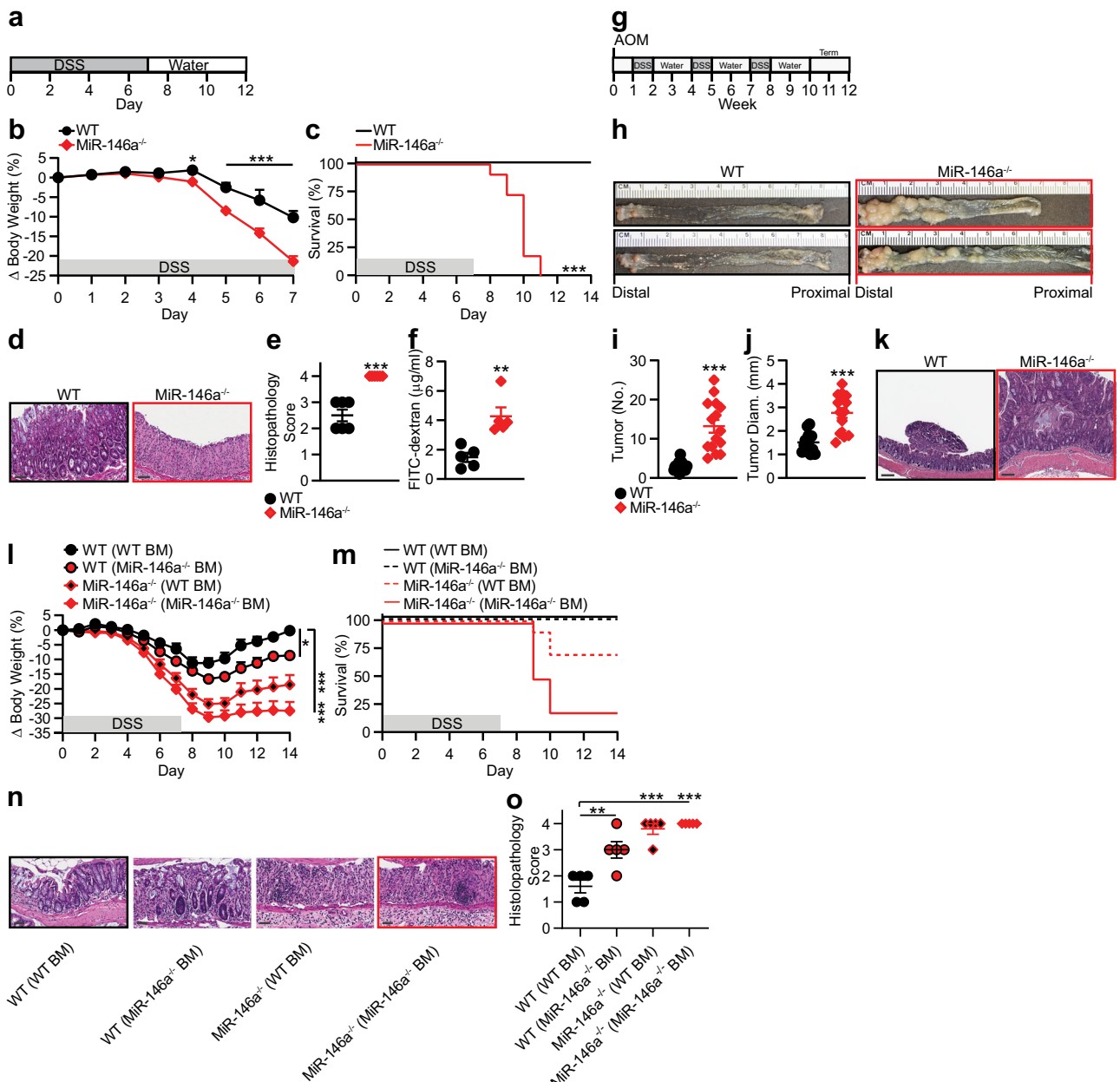

**Fig. 1 MiR-146a deficiency promotes colonic inflammation and colitis-associated colon cancer. a** Schematic of DSS colitis induction: 3% DSS was given in water for 7 days, followed by untreated drinking water for 7 days. **b, c** Percent body weight changes ($n = 15$) (**b**) and survival ($n = 11$) (**c**) in colitic WT and miR-146a$^{-/-}$ mice. **d, e** Representative colonic histopathology with H&E staining (scale bar = 50 μm) (**d**) and histopathological scores (**e**) based on degree of ulceration from colitic mice on day 7 ($n = 6$). **f** FITC-dextran uptake in colitic mice on day 7. Mice were deprived of food and water for 6 h, then given FITC-dextran (400 mg/kg b.w.) by oral gavage. Four hours later, mice were bled and serum was collected. Fluorescence intensity was measured by spectrophotometer ($n = 5$). **g** Schematic of AOM/DSS inflammation-associated CRC induction: AOM was given i.p. (10 mg/kg b.w.), followed by 3 cycles of 2% DSS. **h–j** Representative images (**h**), numbers (**i**), and sizes (**j**) of colonic tumors in WT and miR-146a$^{-/-}$ mice with AOM/DSS-induced CRC ($n = 14$–15). **k** Representative colonic histopathology (scale bar = 200 μm) with H&E staining in WT and miR-146a$^{-/-}$ mice with AOM/DSS-induced CRC. **l, m** Percent body weight changes (**l**) and survival (**m**) in BM chimeric mice with DSS-induced colitis. WT and miR-146a$^{-/-}$ mice were irradiated and reconstituted with WT or miR-146a$^{-/-}$ BM, for 6 weeks ($n = 9$–10). Weight changes were compared to WT (WT BM) on day 14. **n, o** Representative colonic histopathology (scale bar = 50 μm) stained with H&E (**n**) and histopathological scores (**o**) of colitic chimera mice on day 7 ($n = 5$). Comparisons are to WT (WT BM). Data are representative of ≥2 independent experiments. $n$ = biologically independent replicates per group. Mean ± SEM. *$p < 0.05$, **$p < 0.01$, ***$p < 0.001$, by two-way ANOVA with Bonferroni adjustment (**b**), two-way Kaplan–Meier survival analysis (**c**), one-way ANOVA with Dunnett adjustment (**l, o**), or two-tailed Student's $t$-test (**e, f, i, j**). Source data are provided as a Source data file.

CRC in miR-146a$^{-/-}$ mice could be linked to enhanced IL-17 and IL-17-promoting cytokines by myeloid cells.

Myeloid cells, such as MΦs and DCs, play a critical role in CRC development, including the promotion of tumorigenic IL-17 in the colonic microenvironment[31,32]. As we found that inflammation-associated CRC susceptibility in global miR-146a$^{-/-}$ mice was associated with enhanced inflammatory myeloid cell gene signatures, we tested the impact of myeloid cell-intrinsic miR-146a in limiting CRC. For this, we generated mice with a conditional deletion of miR-146a in the myeloid compartment[33,34]: LysM$^{Cre}$miR-146a$^{fl/fl}$

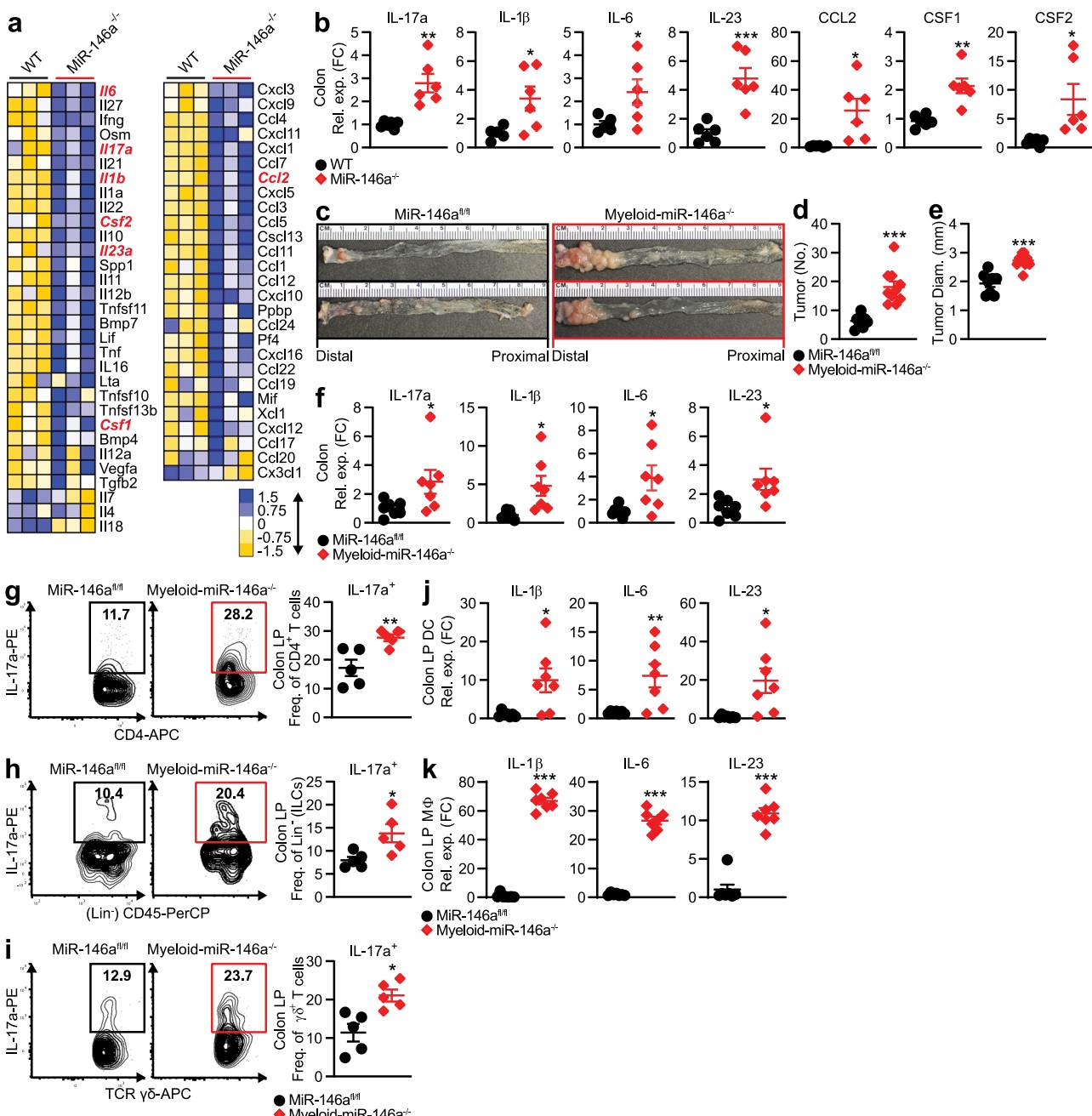

**Fig. 2 Myeloid cell-specific deletion of miR-146a promotes CRC development. a** Inflammatory Cytokines and Receptors RT2 Profiler PCR Array (Qiagen) in distal colon from colitic WT and miR-146a$^{-/-}$ mice ($n = 3$). **b** qPCR of IL-17 and IL-17-promoting cytokines in CRC tissue from WT and miR-146a$^{-/-}$ mice with AOM/DSS-induced CRC ($n = 6$-7). **c–e** Representative images (**c**), numbers (**d**), and sizes (**e**) of colonic tumors in miR-146a$^{fl/fl}$ and myeloid-miR-146a$^{-/-}$ (LysM$^{Cre}$miR-146a$^{fl/fl}$) mice with AOM/DSS-induced CRC ($n = 8$-10). **f** qPCR of IL-17 and IL-17-inducing cytokines in CRC tissue from miR-146a$^{fl/fl}$ and myeloid-miR-146a$^{-/-}$ mice ($n = 7$). **g–i** Representative FACS plots and frequencies of IL-17a in CD4$^+$ T cells (Th17) (**g**), (Lin$^-$CD45$^+$) ILCs (**h**), and (CD3$^+$ δ$^+$) γδ$^+$ T cells (**i**) from the colonic LP of miR-146a$^{fl/fl}$ and myeloid-miR-146a$^{-/-}$ mice with AOM/DSS-induced CRC ($n = 5$). **j, k** qPCR of IL-17-inducing cytokines in CD11c$^+$ DCs (**j**) and CD11b$^+$ MΦs (**k**) from the colonic LP of CRC miR-146a$^{fl/fl}$ and myeloid-miR-146a$^{-/-}$ mice ($n = 7$). qPCR data as FC from WT or miR-146a$^{fl/fl}$ (**b, f, j, k**). Data are representative of ≥2 independent experiments. $n$ = biologically independent replicates per group. Mean ± SEM. *$p < 0.05$, **$p < 0.01$, ***$p < 0.001$, by two-tailed Student's $t$-test (**b, d–k**). Source data are provided as a Source data file.

(myeloid-miR-146a$^{-/-}$—myeloid cell miR-146a conditional knockout). We found this myeloid-specific deletion of miR-146a leads to worsened colitis (Supplementary Fig. 2a, b) and colitis-associated CRC (Fig. 2c–e), as seen in global miR-146a$^{-/-}$ mice (Fig. 1h–j). The susceptibility to CRC in myeloid-miR-146a$^{-/-}$ mice was associated with increased levels of IL-17 and IL-17-promoting cytokines in CRC tissues (Fig. 2f). As CD4$^+$ Th17 cells, γδ$^+$ T cells, and ILCs are major IL-17 producers in the tumor microenvironment in the gut[15], we

investigated the cellular source of IL-17, which contributed to the overall increase in IL-17 levels seen in myeloid-miR-146a$^{-/-}$ mice during CRC. We found enhanced IL-17 production by CD4$^+$ Th17 cells, γδ$^+$ T cells, and ILCs in the colonic lamina propria (LP) of CRC myeloid-miR-146a$^{-/-}$ mice compared to miR-146a$^{fl/fl}$ mice with intact miR-146a expression (Fig. 2g–i). A similar increase in IL-17 was also observed in global miR-146a$^{-/-}$ mice with CRC (Supplementary Fig. 2c–e). IL-17 in these populations is mainly

induced by myeloid cells, such as DCs and MΦs, which respond to microbial products by releasing IL-17-promoting cytokines[16]. In the gut, DCs appear to represent a particularly important source of IL-17-promoting cytokines[17]. Our analyses of sorted myeloid cells from the colonic LP[35] (Supplementary Fig. 2f) of tumor-bearing myeloid-miR-146a$^{-/-}$ mice (Fig. 2j, k) and global miR-146a$^{-/-}$ mice (Supplementary Fig. 2g) revealed upregulation of several IL-17-promoting cytokines. Together, these data suggested miR-146a-deficiency in myeloid cells promotes the tumorigenic secretion of IL-17 and confers susceptibility to CRC.

**Myeloid-miR-146a limits RIPK2 and IL-17-promoting cytokines**. Next, we investigated the molecular mechanism by which miR-146a within myeloid cells limits IL-17-promoting cytokines and IL-17 producers in CRC. Among the IL-17-promoting cytokines, IL-23 is indispensable for the expansion of IL-17 producers in the colon[17]. Recent work has shown that NOD2 signaling in DCs is critical for the secretion of IL-17-inducing cytokines such as IL-23 and promotes the induction of colonic IL-17[17,18]. To explore whether increased IL-17-promoting cytokines in miR-146a$^{-/-}$ mice are associated with enhanced NOD2 signaling, we measured receptor interacting serine/threonine kinase 2 (RIPK2), an essential adaptor molecule implicated in NOD2 signaling[36]. We found increased levels of RIPK2 in CRC tissues (Fig. 3a) and colonic LP DCs and MΦs (Fig. 3b, c) from myeloid-miR-146a$^{-/-}$ mice. A similar increase in RIPK2 in CRC tissues and colonic LP myeloid cells was also observed in global miR-146a$^{-/-}$ mice (Supplementary Fig. 3a, b). These findings subsequently prompted us to test whether miR-146a limits NOD2 signaling by targeting RIPK2 in myeloid cells. Interestingly, our in silico analysis using the RNAhybrid tool[37] predicted RIPK2 as a direct target of miR-146a (Fig. 3d). To determine whether RIPK2 was a bona fide target of miR-146a in myeloid cells, we performed luciferase assays in a MΦ cell line (RAW 264.7) transfected with a luciferase construct containing RIPK2 3′-UTR in the presence of control or miR-146a mimic. We found that miR-146a inhibited the luciferase activity of a reporter containing RIPK2 3′-UTR (Fig. 3e). We also validated direct binding of miR-146a to RIPK2 by streptavidin pulldown from RAW 264.7 cells transfected with biotinylated miR-146a mimic. We observed robust enrichment for RIPK2 in miR-146a-pulldown lysates from miR-146 mimic-transfected cells (Fig. 3f). In addition, we detected RIPK2 downregulation in miR-146-overexpressing RAW 264.7 cells (Fig. 3g). Together, these data indicated that RIPK2 is a direct target of miR-146a.

The effect of miR-146a deficiency on RIPK2-mediated NOD2 signaling was then examined in miR-146a$^{-/-}$ myeloid cells. MiR-146a$^{-/-}$ DCs expressed higher levels of RIPK2 (Fig. 3h) and showed stronger activation of RIPK2 downstream targets, NF-κB [phospho-p65 (pp65)], and phospho-p38 MAPK (pp38) (Fig. 3i, j), when activated with the NOD2 ligand muramyl dipeptide (MDP). We also observed a similar increase in both RIPK2 and downstream NOD2 signaling in miR-146a$^{-/-}$ MΦs stimulated with MDP (Fig. 3k–m). Accordingly, levels of IL-17-promoting cytokines, including IL-23, were elevated upon MDP stimulation in DCs and MΦs from miR-146a$^{-/-}$ mice (Fig. 3n, o). It is important to consider that in MΦs and particularly in DCs, LysM-Cre-mediated genetic deletion has been reported to be incomplete[33,34]. Thus, we sought to confirm that the enhanced RIPK2 and IL-17-promoting cytokine expression we observed in myeloid cells from CRC myeloid-miR-146a$^{-/-}$ mice (Figs. 2j, k) was due to the direct effect of miR-146a loss within these cell types. Consistent with our in vivo data (Figs. 2j, k and 3b, c), DCs and MΦs isolated from myeloid-miR-146a$^{-/-}$ mice also showed enhanced RIPK2 and IL-17-promoting cytokine expression in

response to MDP stimulation in vitro (Supplementary Fig. 3c–f). These data further emphasized the critical role of fully intact miR-146a in limiting IL-17-promoting responses from myeloid cells.

Next, we explored whether miR-146a targeting of RIPK2 drives enhanced IL-17-promoting cytokine induction in miR-146a-deficient myeloid cells. Similar to other pattern-recognition receptors, NOD2 has also been shown to signal via the classical (canonical) NF-κB pathway downstream of RIPK2[38]. MiR-146a has been reported to target other canonical and noncanonical NF-κB subunits, c-Rel and RelB, as well as an NF-κB signaling molecule downstream of RIPK2, IKKα, in myeloid cells and B cells[39,40]. We measured the expression of these molecules in miR-146a$^{-/-}$ myeloid cells and found some moderate increases at protein levels alongside RIPK2 in miR-146a$^{-/-}$ myeloid cells (Supplementary Fig. 3g, h). However, pharmacologic inhibition of RIPK2 alone strikingly abrogated MDP-triggered IL-17-promoting cytokines in miR-146a$^{-/-}$ DCs and MΦs, abolishing differences between WT and miR-146a$^{-/-}$ cells (Fig. 3p, q). This finding appeared to reflect the indispensable function of the miR-146a-RIPK2 axis as an upstream regulator of canonical NF-κB signaling, which plays a major role in IL-17-promoting cytokine secretion[17,18].

Because DCs, as professional antigen-presenting cells, are major regulators of IL-17 production in the gut[17], we next asked whether miR-146a$^{-/-}$ DCs could directly promote IL-17 production. Consistent with elevated IL-17-promoting cytokines from miR-146a$^{-/-}$ myeloid cells (Fig. 3n, o), increased IL-17 induction was observed in CD4$^+$ T cells cocultured with miR-146a$^{-/-}$ DCs (Fig. 3r). A similar increase in IL-17 induction was also observed in γδ$^+$ T cells and ILCs cocultured with miR-146a$^{-/-}$ DCs (Supplementary Fig. 3i, j). These data further implicated a critical role for miR-146a expression within myeloid cells in limiting IL-17 levels during CRC (Fig. 2f–i). Indeed, we found that blocking IL-17 during CRC could ameliorate increased CRC severity in myeloid-miR-146a$^{-/-}$ mice (Fig. 3s, t). Together, these findings suggested that miR-146a limits IL-17-promoting cytokines in myeloid cells by targeting RIPK2, thereby inhibiting myeloid-driven IL-17 producers in the colonic microenvironment and reducing CRC development.

**IEC-specific deletion of miR-146a promotes CRC**. CRC develops in IECs lining the colon and rectum[41]. Our BM chimera experiment (Fig. 1l–o) suggested a role for miR-146a expressed in non-hematopoietic cells (possibly IECs) for preventing colonic inflammation and CRC development. To investigate whether IEC-intrinsic miR-146a modulates CRC development, we generated mice with a conditional deletion of miR-146a in IECs:[42] Villin$^{Cre}$miR-146a$^{fl/fl}$ (IEC-miR-146a$^{-/-}$—IEC-miR-146a conditional knockout). We found IEC-miR-146a$^{-/-}$ mice were susceptible to DSS-induced colitis (Supplementary Fig. 4a, b). Accordingly, when IEC-miR-146a$^{-/-}$ mice were subjected to an AOM/DSS model of CRC, they displayed a marked increase in the number of macroscopically visible tumors (Fig. 4a–c), supporting the idea that miR-146a may also function within IECs to prevent colonic inflammation and CRC development.

IEC expression of IL-17R is critical for the development of CRC[7]. IL-17R signaling in IECs activates NF-κB and MAPKs, both of which promote survival pathways required for the growth of premalignant lesions and subsequent development of tumors[7]. In fact, IEC-specific deletion of IL-17R is known to prevent IL-17-mediated tumor development[7]. To investigate whether miR-146a functions within IECs to limit tumorigenic IL-17R signaling, we profiled molecules associated with IL-17R signaling in CRC tissues from IEC-miR-146a$^{-/-}$ mice. We found a dramatic increase in TNF receptor-associated factor 6 (TRAF6), an adaptor molecule and

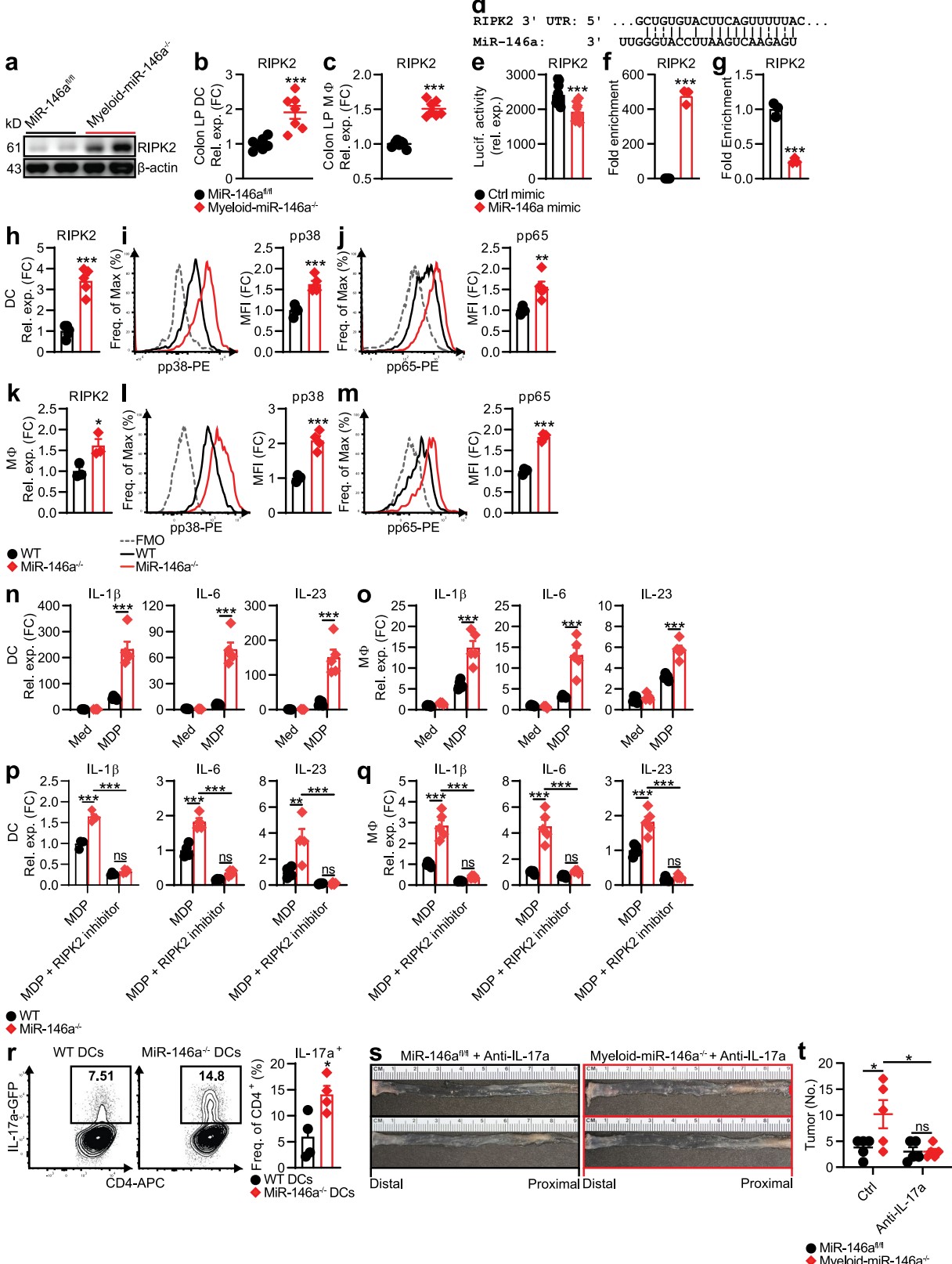

initial signaling intermediate necessary for IL-17-induced activation of NF-κB and MAPKs[43] (Fig. 4d). Accordingly, we also found an increased activation of NF-κB [phospho-p65 (pp65)] and MAPKs family member, pp38, in CRC tissues from miR-146a$^{-/-}$ mice (Fig. 4d). A similar increase in IL-17R signaling was also detected in

CRC tissues from global miR-146a$^{-/-}$ mice (Supplementary Fig. 4c). However, in contrast to global miR-146a$^{-/-}$ mice (Fig. 2b and Supplementary Fig. 2c), we found no increase in IL-17 expression in CRC tissues or within IL-17-producing immune cells in IEC-miR-146a$^{-/-}$ mice (Supplementary Fig. 4d, e). Therefore, we explored the

**Fig. 3 Myeloid cell-specific deletion of miR-146a enhances tumorigenic IL-17-promoting cytokines by promoting NOD2-RIPK2 signaling. a** Western blotting of RIPK2 in CRC tissue from miR-146a$^{fl/fl}$ and myeloid-miR-146a$^{-/-}$ mice. **b, c** qPCR of RIPK2 in CD11c$^+$ DCs (**b**) and CD11b$^+$ MΦs (**c**) from the colonic LP of these CRC mice ($n = 7$). **d** In silico target prediction of miR-146a binding the RIPK2 3′-UTR. **e** Luciferase reporter activity in macrophage cell line (RAW 264.7) transfected with luciferase construct containing RIPK2 sequence with ctrl or miR-146a mimic ($n = 10$). **f** qPCR of RIPK2 from streptavidin pulldown of biotinylated ctrl or miR-146a mimic-transfected RAW 264.7 cells ($n = 3$). **g** qPCR of RIPK2 in RAW 264.7 cells transfected with ctrl or miR-146a mimic ($n = 3$). **h** qPCR of RIPK2 in WT and miR-146a$^{-/-}$ BM-derived DCs stimulated with MDP (10 μg/ml) for 24 h ($n = 5$). **i, j** Representative FACS histograms and MFIs of phospho-p38 MAPK (**i**) and NF-κB subunit, phospho-p65 (**j**) in these BMDCs stimulated with MDP (10 μg/ml) for 1.5 h ($n = 5$). **k** qPCR of RIPK2 in WT and miR-146a$^{-/-}$ BM-derived MΦs stimulated with MDP (10 μg/ml) for 24 h ($n = 3$). **l, m** Representative FACS histograms and MFIs of phospho-p38 MAPK (**l**) and phospho-p65 (**m**) in these BMDMs stimulated with MDP (10 μg/ml) for 1.5 h ($n = 4$). **n, o** qPCR of IL-17-inducing cytokines in WT and miR-146a$^{-/-}$ BMDCs (**n**) and BMDMs (**o**) stimulated with MDP (10 μg/ml) for 24 h ($n = 5$). **p, q** qPCR of IL-17-inducing cytokines in these BMDCs (**p**) ($n = 4$) and BMDMs ($n = 5$) (**q**) stimulated with MDP (10 μg/ml) in the presence or absence of RIPK2 inhibitor (500 nM) for 24 h. **r** Representative FACS plots and frequencies of IL-17a in IL-17$^{GFP}$ CD4$^+$ T cells cocultured with WT or miR-146a$^{-/-}$ BMDCs at a 1:1 T cell:DC ratio for 5 days with soluble anti-CD3 and anti-CD28 (1 μg/ml), under low-dose Th17-polarizing conditions with TGF-β (0.5 ng/ml), IL-6 (10 ng/ml), and anti-IFN-γ (10 μg/ml). BMDCs were prestimulated with MDP for 20 h, then washed before coculture ($n = 4$). **s, t** Representative images (**s**) and numbers (**t**) of colonic tumors in miR-146a$^{fl/fl}$ and myeloid-miR-146a$^{-/-}$ mice treated with anti-IL-17a (500 μg/mouse) twice a week throughout AOM/DSS CRC induction ($n = 5$). qPCR data as FC from miR-146a$^{fl/fl}$ (**b, c**), WT MDP (**h, k, p, q**), or WT med (**n, o**). MFI data as FC from WT with Fluorescence Minus One (FMO) ctrls shown (**i, j, l, m**). Data are representative of ≥2 independent experiments. $n =$ biologically independent replicates per group (**b, c, h-r, t**) or replicates pooled from independent experiments (**e-g**). Mean ± SEM. *$p < 0.05$, **$p < 0.01$, ***$p < 0.001$, by two-way ANOVA with Tukey adjustment (**n-q, t**) or two-tailed Student's $t$-test (**b, c, e-m, r**). Source data are provided as a Source data file.

possibility that miR-146a might function within IECs to directly limit tumorigenic IL-17R signaling and responsiveness to IL-17.

Our in silico analysis using the web-based target prediction software program, TargetScan, predicted TRAF6 as a direct target of miR-146a (Fig. 4e). In fact, recent data from other disease models suggest that miR-146a is an upstream regulator of TRAF6[44]. To determine whether miR-146a directly targets TRAF6 in IECs to limit IL-17R signaling, we performed luciferase assays in an IEC line (CMT-93) transfected with a luciferase construct containing TRAF6 3′-UTR in the presence of control or miR-146a mimic. We found that miR-146a inhibited the luciferase activity of a reporter containing TRAF6 3′-UTR (Fig. 4f). We also validated direct binding of miR-146a to TRAF6 by streptavidin pulldown from IECs transfected with biotinylated miR-146a mimics, in which miR-146a pulldown lysates were robustly enriched with TRAF6 (Fig. 4g). Consistent with this, colonic IECs[45] sorted (Supplementary Fig. 4f) from CRC IEC-miR-146a$^{-/-}$ mice (Fig. 4h) and global miR-146a$^{-/-}$ mice (Supplementary Fig. 4c) expressed higher levels of TRAF6 compared to WT mice. We also confirmed corresponding increases in downstream IL-17R signaling directly within IECs from CRC IEC-miR-146a$^{-/-}$ mice (Fig. 4i) and global miR-146a$^{-/-}$ mice (Supplementary Fig. 4h), as we had observed in CRC tissues from CRC IEC-miR-146a$^{-/-}$ mice (Fig. 4d) and global miR-146a$^{-/-}$ mice (Supplementary Fig. 4c). To functionally test whether miR-146a limits tumorigenic IL-17R signaling within IECs, we generated stable CMT-93 IEC lines that were sufficient or deficient in miR-146a (Supplementary Fig. 4i). These miR-146a-sufficient (WT) and miR-146a-deficient ($^{-/-}$) IECs were then stimulated with IL-17 and analyzed for activation of IL-17R signaling intermediates. We found that miR-146a$^{-/-}$ IECs expressed higher levels of TRAF6 and respond robustly to IL-17 stimulation through NF-κB and p38 MAPK activation compared to WT IECs (Fig. 4j). These results suggested that miR-146a within IECs limits IL-17R signaling and subsequent CRC development by targeting TRAF6, thereby inhibiting downstream NF-κB and MAPKs activation.

Among the inflammatory mediators implicated in CRC growth and progression, prostaglandin E2 (PGE2) plays a key role[46]. Elevated levels of PGE2 and cyclooxygenase-2 (Cox-2), an upstream inducer of PGE2 synthesis, have been found in a majority of adenocarcinomas and are associated with worse survival in CRC patients[46]. Cancerous epithelial cells have been shown to be a major producer of PGE2 in the colonic microenvironment[46]. Interestingly, IL-17 has recently been shown to induce the upregulation of Cox-2 and subsequent production of PGE2 in cancer cells in vitro via the NF-κB pathway[47]. As we found that CRC susceptibility in IEC-miR-146a$^{-/-}$ mice is associated with enhanced IL-17R signaling, including activation of NF-κB, we tested whether this enhanced IL-17R signaling also led to increased Cox-2-PGE2 signaling. Intriguingly, we detected increased levels of Cox-2 and β-catenin, a downstream PGE2 target, in CRC tissues and sorted IECs from IEC-miR-146a$^{-/-}$ mice (Fig. 4k, l) and miR-146a$^{-/-}$ mice (Supplementary Fig. 4j, k). Directly stimulating miR-146a$^{-/-}$ IECs in vitro with IL-17 also resulted in enhanced Cox-2 and β-catenin (Fig. 4m). In addition, we found that this increase in Cox-2 and β-catenin in miR-146a$^{-/-}$ IECs was associated with increased PGE2 levels (Fig. 4n). These results suggested that miR-146a limits tumorigenic PGE2 levels within IECs by targeting TRAF6, thereby limiting IL-17-induced Cox-2.

Interestingly, our analyses of CRC tissues and IECs from IEC-miR-146a$^{-/-}$ mice (Fig. 4o-r) and miR-146a$^{-/-}$ mice (Supplementary Fig. 4l-o) revealed higher RNA and protein levels of prostaglandin E synthase 2 (PTGES2), a major enzyme involved in PGE2 synthesis, which converts PGH2 to PGE2[48]. PTGES2 levels were also increased in miR-146a$^{-/-}$ IECs directly stimulated with IL-17 (Fig. 4s). These findings led us to test whether miR-146a directly regulated PGE2 by targeting PTGES2 (in addition to indirectly regulating PGE2 by limiting IL-17R-induced TRAF6 and Cox-2). In agreement with a report on neuronal stem cells[49], our in silico analysis using the RNAhybrid tool[37] suggested that miR-146a binds to PTGES2 (Fig. 4t). To determine whether miR-146a targets PTGES2 in IECs to limit PGE2 levels, we performed luciferase assays in an IEC line (CMT-93) transfected with a luciferase construct containing PTGES2 3′-UTR in the presence of control or miR-146a mimic. We found that miR-146a inhibited the luciferase activity of a reporter containing PTGES2 3′-UTR (Fig. 4u). We also validated miR-146a binding to PTGES2 by streptavidin pulldown from IECs transfected with biotinylated miR-146a mimics (Fig. 4v). These findings indicated that PTGES2 was a direct target of miR-146a. Consistent with increased IL-17-mediated tumorigenic signaling, including PGE2 in IECs, we found miR-146a$^{-/-}$ CRC tissues expressed higher levels of Ki67, a marker of proliferation (Fig. 4w). Finally, we demonstrated that blocking IL-17 signaling by IL-17 neutralization during CRC could ameliorate the enhanced CRC severity we observed in IEC-miR-146a$^{-/-}$ mice (Fig. 4x, y) and global miR-146a$^{-/-}$ mice (Supplementary Fig. 4p, q). Altogether, these data suggested that miR-146a limits tumorigenic IL-17R signaling in IECs by three

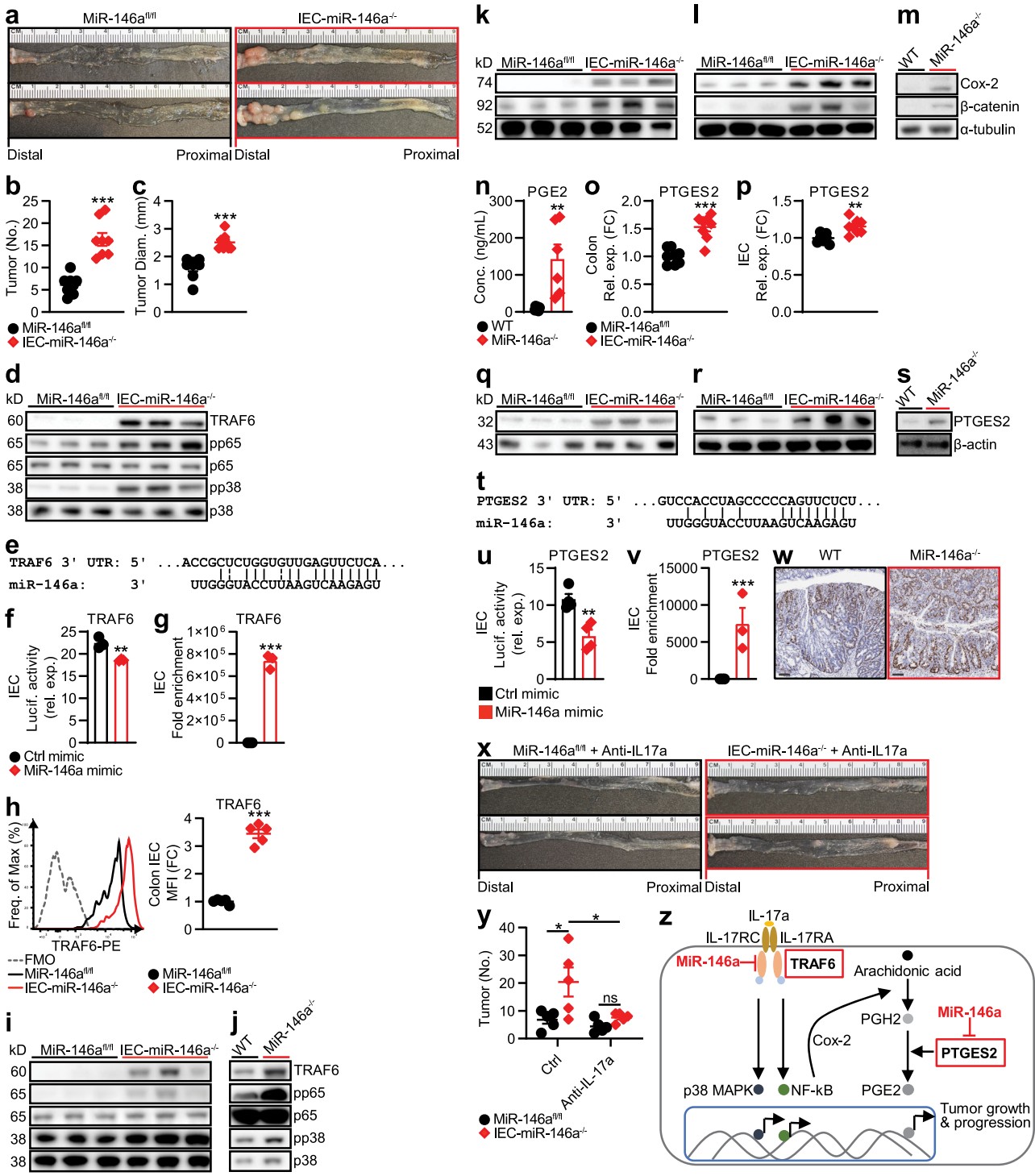

interlinked mechanisms: (1) preventing NF-κB and p38 MAPK pathways by targeting TRAF6; (2) limiting PGE2 levels by targeting IL-17R-TRAF6-mediated induction of Cox-2; and (3) directly targeting PTGES2, an enzyme that converts PGH2 to PGE2[48], to confer protection against CRC (Fig. 4z). Our results support an IEC-intrinsic effect of miR-146a-deficiency leading to enhanced CRC development.

**MiR-146a deletion promotes spontaneous tumors in Apc$^{Min/+}$ mice.** Emerging evidence suggests that persistent inflammation, in particular IL-17, not only promotes colitis-associated CRC but

also sporadic CRC[1,2]. Elevated levels of IL-17 have been observed in 80% of human sporadic cancer tissues[50] and ablation of IL-17 can inhibit the progression of spontaneous intestinal tumorigenesis in Apc$^{Min/+}$ mice[5]. Importantly, altered levels of miR-146a are linked to spontaneous CRC outcomes in humans[23,26]. As we found miR-146a suppressed inflammation-associated CRC by limiting both IL-17 production and IL-17-mediated tumorigenic signaling, we asked whether miR-146a also exerted its protective effects in spontaneous CRC. To address this hypothesis, we generated miR-146a$^{-/-}$ mice that contained a heterozygous mutation in the *adenomatous polyposis coli* (*Apc*) gene

**Fig. 4 IEC-specific deletion of miR-146a promotes CRC development. a–c** Representative images (**a**), numbers (**b**), and sizes (**c**) of colonic tumors in miR-146a^fl/fl and IEC-miR-146a^−/− (Villin^Cre^MiR-146a^fl/fl) mice with AOM/DSS-induced CRC (*n* = 8). **d** Western blottings of TRAF6, NF-κB subunit phospho-p65 (pp65), and phospho-p38 (pp38) MAPK in CRC tissue from these mice. **e** In silico target prediction of miR-146a binding the TRAF6 3′-UTR. **f** Luciferase activity in IEC line (CMT-93) transfected with luciferase construct containing TRAF6 sequence with ctrl or miR-146a mimic (*n* = 3). **g** qPCR of TRAF6 from streptavidin pulldown of biotinylated ctrl or miR-146a mimic-transfected CMT-93 IECs (*n* = 3). **h** Representative FACS histograms and MFIs of TRAF6 in FACS-sorted IECs from miR-146a^fl/fl and IEC-miR-146a^−/− mice with AOM/DSS-induced CRC. MFI as FC from miR-146a^fl/fl (*n* = 5). **i** Western blottings of TRAF6, pp65, and pp38 in IECs from these CRC mice. **j** Western blottings of these molecules in miR-146a-sufficient (WT) and miR-146a-deficient (miR-146a^−/−) CMT-93 IECs stimulated with IL-17 (25 ng/ml) for 1.5 h. **k, l** Western blottings of Cox-2 and β-catenin in CRC tissue (**k**) and IECs (**l**) from CRC miR-146a^fl/fl and IEC-miR-146a^−/− mice. **m** Western blottings of Cox-2 and β-catenin in WT and miR-146a^−/− CMT-93 IECs stimulated with IL-17a (25 ng/ml). **n** ELISA of PGE2 in WT and miR-146a^−/− CMT-93 IECs stimulated with IL-17a (25 ng/ml) for 48 h (*n* = 6). **o, p** qPCR of PTGES2 in CRC tissue (**o**) and IECs (**p**) from CRC miR-146a^fl/fl and IEC-miR-146a^−/− mice. qPCR data as FC from miR-146a^fl/fl (*n* = 8). **q, r** Western blotting of PTGES2 in CRC tissue (**q**) and IECs (**r**) from CRC miR-146a^fl/fl and IEC-miR-146a^−/− mice. **s** Western blotting of PTGES2 in WT and miR-146a^−/− CMT-93 IECs stimulated with IL-17a (25 ng/ml). **t** In silico target prediction of miR-146a binding the PTGES2 3′-UTR. **u** Luciferase activity in CMT-93 IECs transfected with luciferase construct containing PTGES2 sequence with ctrl or miR-146a mimic (*n* = 4). **v** qPCR of PTGES2 from streptavidin pulldown of biotinylated ctrl or miR-146a mimic-transfected CMT-93 IECS (*n* = 3). **w** Representative colonic histopathology (scale bar = 100 μm) with Ki67 staining in WT and miR-146a^−/− mice with AOM/DSS-induced CRC. **x, y** Representative images (**x**) and numbers (**y**) of colonic tumors in miR-146a^fl/fl and IEC-miR-146a^−/− mice treated with anti-IL-17a (500 μg/ml) twice a week throughout AOM/DSS CRC induction (*n* = 5). **z** Model depicting IEC-intrinsic miR-146a in limiting tumorigenic IL-17 signaling in IECs. Data are representative of ≥2 independent experiments. *n* = biologically independent replicates per group (**b, c, h, o, p, y**) or replicates pooled from independent experiments (**f, g, n, u, v**). Mean ± SEM. *$p < 0.05$, **$p < 0.01$, ***$p < 0.001$, by two-way ANOVA with Tukey's adjustment (**y**) or two-tailed Student's *t*-test (**b, c, f–h, n–p, u, v**). Source data are provided as a Source data file.

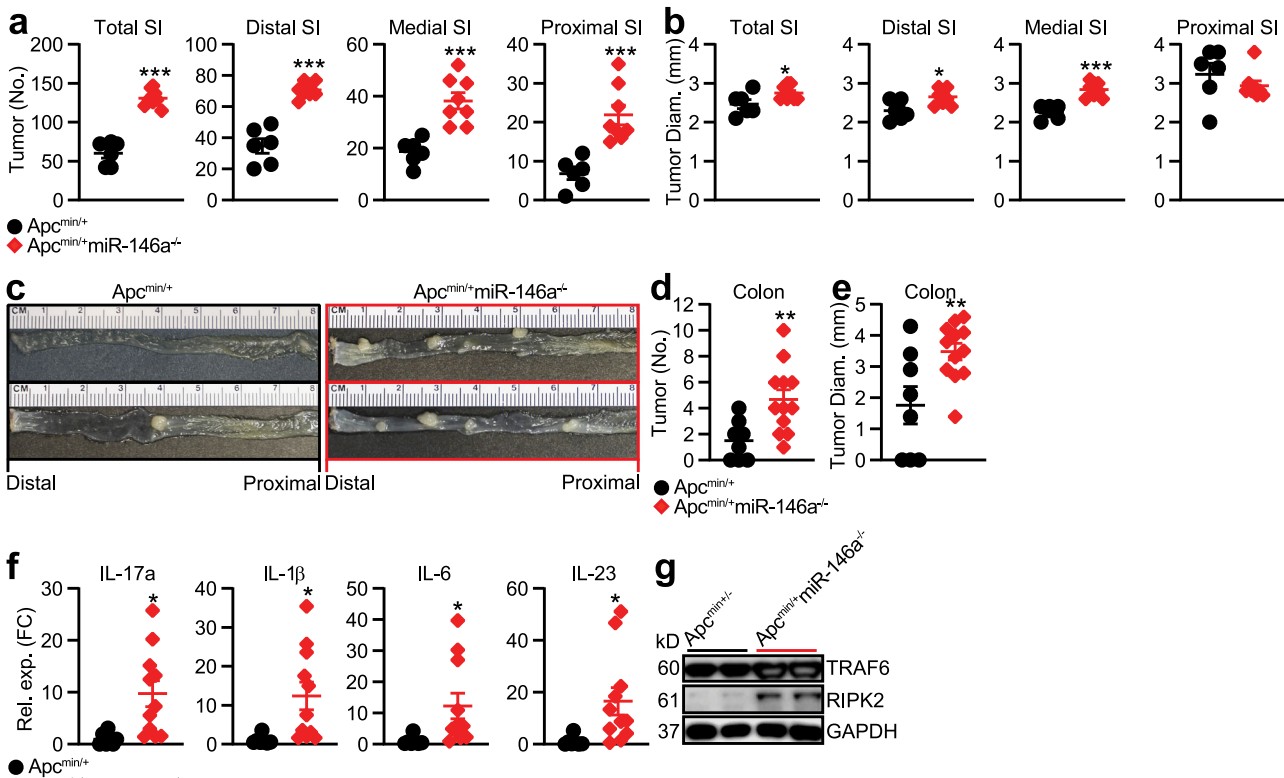

**Fig. 5 MiR-146a deficiency promotes spontaneous tumor development in Apc^Min/+ mice. a, b** Small Intestine (SI) tumor numbers (**a**) and sizes (**b**) of Apc^Min/+ and Apc^Min/+miR-146a^−/− mice at 22–24 weeks of age (*n* = 6–8). **c–e** Representative images (**c**), numbers (**d**), and sizes (**e**) of colonic tumors in Apc^Min/+ and Apc^Min/+miR-146a^−/− mice at 22–24 weeks of age (*n* = 8–12). **f** qPCR of IL-17 and IL-17-inducing cytokines in colonic tumor tissue from these mice. qPCR data as FC from Apc^Min/+ (*n* = 8–12). **g** Western blottings of TRAF6 and RIPK2 in CRC tissue from these mice. Data are representative of ≥2 independent experiments. *n* = biologically independent replicates per group. Mean ± SEM. *$p < 0.05$, **$p < 0.01$, ***$p < 0.001$, by two-tailed Student's *t*-test (**a, b, d–f**). Source data are provided as a Source data file.

(Apc^Min/+miR-146a^−/−), as mutations in *Apc* are responsible for driving sporadic CRC in humans[51]. We observed Apc^Min/+miR-146a^−/− mice developed greater numbers of tumors in both the small intestine (Fig. 5a, b) and colon (Fig. 5c–e) compared to that in WT Apc^Min/+ mice. These data demonstrated that miR-146a also restricts spontaneous CRC.

To further investigate whether the susceptibility to spontaneous tumorigenesis in Apc^Min/+miR-146a^−/− mice was also associated with enhanced IL-17 signaling, we profiled the expression of IL-17 and IL-17-promoting cytokines in colonic homogenates. We detected elevated levels of IL-17 and IL-17-promoting cytokines such as IL-1β, IL-6, and IL-23

in Apc$^{Min/+}$miR-146a$^{-/-}$ mice compared to WT Apc$^{Min/+}$ mice (Fig. 5f). In addition, we found that the increased IL-17 signaling in Apc$^{Min/+}$miR-146a$^{-/-}$ mice was associated with an increase in myeloid- and IEC-miR-146a targets, RIPK2 and TRAF6 (Fig. 5g). These results suggested that miR-146a also limits tumorigenic IL-17 signaling in sporadic intestinal tumors of Apc$^{Min/+}$ mice.

**MiR-146a mimic or target inhibition therapy ameliorates CRC.**
As we identified that miR-146a deficiency promotes colonic inflammation and CRC, we tested whether overexpression of miR-146a could ameliorate colonic inflammation. We found miR-146a mimic treatment (Fig. 6a) effectively ameliorated colitis (Fig. 6b). Specifically, miR-146a mimic-treated mice lost less weight and displayed less colonic inflammation (Fig. 6b–d). We

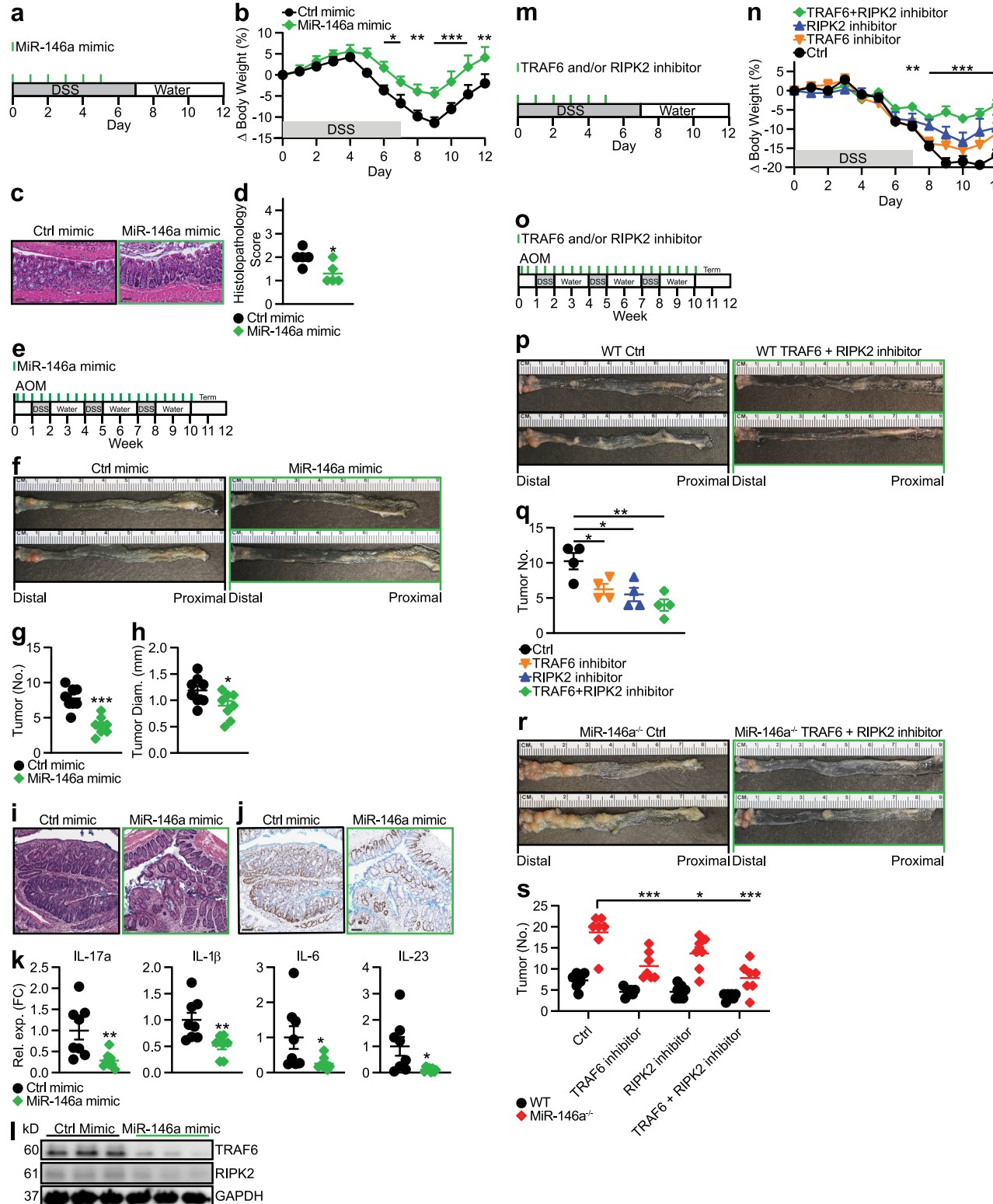

**Fig. 6 MiR-146a mimic therapy and miR-146a target inhibition therapy ameliorate colonic inflammation and colitis-associated CRC. a** Schematic of miR-146a mimic treatment during DSS (3%)-induced colitis. Ctrl or miR-146a mimic (5 mg/kg b.w.) in in vivo RNA transfection reagent was given i.p. to WT mice. **b** Percent body weight changes of ctrl and miR-146a mimic-treated mice (n = 5). **c, d** Representative colonic histopathology (scale bar = 50 μm) stained with H&E (**c**) and histopathological scores (**d**) of treated mice (n = 5). **e,** Schematic of miR-146a mimic treatment paradigm during AOM/DSS-induced CRC. Ctrl or miR-146a mimic (5 mg/kg b.w.) in in vivo RNA transfection reagent was given i.p. to WT mice. **f–h** Representative images (**f**), numbers (**g**), and sizes (**h**) of colonic tumors in treated mice (n = 8). **i, j** Representative colonic histopathology (scale bar = 100 μm) with (**i**) H&E or (**j**) Ki67 staining from treated mice. **k** qPCR of IL-17 and IL-17-inducing cytokines in CRC tissue from treated mice (n = 8). qPCR data as FC from ctrl mimic. **l** Western blottings of TRAF6 and RIPK2 in CRC tissue from treated mice. **m** Schematic of TRAF6 and/or RIPK2 inhibitor treatment during DSS (3%)-induced colitis. Ctrl vehicle, TRAF6 small molecule inhibitor (5 mg/kg b.w.), RIPK2 small molecule inhibitor (5 mg/kg b.w.), or TRAF6 + RIPK2 small molecule inhibitor combination, in PBS were given i.p. to WT mice. **n** Percent body weight changes of treated mice. Comparisons between ctrl-treated and TRAF6 + RIPK2 inhibitor-treated mice (n = 5). **o** Schematic of TRAF6 and/or RIPK2 inhibitor treatment paradigm during AOM/DSS-induced CRC. Ctrl vehicle, TRAF6 small molecule inhibitor (5 mg/kg b.w.), RIPK2 small molecule inhibitor (5 mg/kg b.w.), or TRAF6 + RIPK2 small molecule inhibitor combination, in PBS were given i.p. to WT mice. **p, q** Representative images (**p**) and numbers (**q**) of colonic tumors in TRAF6 and RIPK2 inhibitor-treated WT mice (n = 4). **r, s** Representative images (**r**) and numbers (**s**) of colonic tumors in treated miR-146a$^{-/-}$ mice (n = 7). Data are representative of ≥2 independent experiments. n = biologically independent replicates per group. Mean ± SEM. *p < 0.05, **p < 0.01, ***p < 0.001, by two-way ANOVA with Bonferroni adjustment (**b, n, s**), two-way ANOVA with Tukey's adjustment (**s**), one-way ANOVA with Tukey adjustment (**q**), or two-tailed Student's t-test (**d, g, h, k, q, s**). Source data are provided as a Source data file.

next tested whether miR-146a mimic therapy could also ameliorate CRC. For this, CRC was induced in WT mice using AOM/DSS and miR-146a mimic or control mimic was administered twice a week for the entire duration of CRC induction (Fig. 6e). We found that miR-146a mimic therapy effectively ameliorated CRC (Fig. 6f–h). Histopathological analyses of colonic sections in miR-146a mimic-treated mice revealed a milder CRC phenotype (Fig. 6i), as well as reduced Ki67 staining (Fig. 6j). In addition, the reduced tumor burden in miR-146a mimic-treated mice was associated with decreased IL-17 and IL-17-promoting cytokines in the colon (Fig. 6k). Furthermore, we measured decreased levels of miR-146a targets, RIPK2 and TRAF6, in CRC tissues from miR-146a mimic-treated mice (Fig. 6l).

Thus far, our data have suggested miR-146a prevents colonic inflammation and CRC development by suppressing both myeloid cell induction of IL-17-producing cells and responsiveness of IECs to IL-17 by targeting RIPK2 in myeloid cells and TRAF6 in IECs, respectively. Next, we explored the therapeutic potential of targeting both RIPK2 and TRAF6 in colitis and colitis-associated CRC. For this, colitic mice were treated with either control, RIPK2 inhibitor, or TRAF6 inhibitor, alone or in combination (Fig. 6m). We found that the combination of both RIPK2 and TRAF6 inhibitors appeared to synergistically ameliorate colitis (Fig. 6n). To investigate whether this synergistic effect applied in the setting of CRC, CRC was induced in WT mice using AOM/DSS and mice were treated with either control, RIPK2 inhibitor, or TRAF6 inhibitor, alone or in combination, twice a week for the entire duration of CRC induction (Fig. 6o). Again, we found that the combination of both RIPK2 and TRAF6 inhibitors appeared to effectively ameliorate CRC (Fig. 6p, q). Finally, we treated CRC-susceptible miR-146a$^{-/-}$ mice with either TRAF6 inhibitor or RIPK2 inhibitor alone, or in combination during AOM/DSS-induced CRC (Fig. 6r, s). Interestingly, direct inhibition of these miR-146a targets were also effective in ameliorating enhanced CRC severity in miR-146a$^{-/-}$ mice (Fig. 6r, s), mice which we have shown express higher levels of RIPK2 and TRAF6 compared to WT. Collectively, these data demonstrated that administration of miR-146a mimic or direct inhibition of the miR-146a targets, TRAF6 and RIPK2, can ameliorate colonic inflammation and CRC.

## Discussion

Although miR-146a has been associated with clinical outcomes in CRC[23,26], our results demonstrate the in vivo function and therapeutic potential of miR-146a in controlling CRC. We have

identified a major protective role for miR-146a in colitis-associated and sporadic colon cancer, including critical cell populations and mechanisms underlying miR-146a-mediated protective functions in CRC. Our data show miR-146a prevents intestinal inflammation and CRC by (1) limiting myeloid cell-mediated inflammatory IL-17 production and by (2) inhibiting tumorigenic IL-17R signaling in IECs (Fig. 7). Mice deficient in miR-146a either globally, specifically within myeloid cells, or specifically within IECs, present with enhanced IL-17 signaling and exacerbated CRC, which is abrogated by IL-17 neutralization. Direct inhibition of miR-146a targets, RIPK2 and TRAF6, also abrogate this susceptibility.

It has been shown that redundant sources of IL-17 production in the gut drive colon tumorigenesis[52]. However, master regulatory mechanisms that limit tumorigenic IL-17 levels in the colonic microenvironment remain unclear. We have identified that miR-146a within myeloid cells has the critical ability to restrict IL-17 production from multiple cell types by inhibiting IL-17-inducing cytokines. Among the myeloid cells, DCs play a major role in promoting IL-17 in the colonic microenvironment[17]. NOD2 signaling in DCs is critical for this secretion of IL-17 promoting cytokines[17]. Our results identify RIPK2 as a previously unknown target of miR-146a in NOD2 signaling within DCs. By targeting RIPK2 in myeloid cells, miR-146a controls NOD2-mediated IL-17-promoting cytokines such as IL-23, which is indispensable for colonic IL-17 induction[17]. As we and others have reviewed, previous work has also implicated miR-146a as a negative regulator of toll-like receptor (TLR)-mediated inflammation[20,44,53–55]. We reveal in the present study another avenue by which miR-146a within myeloid cells can limit IL-17 induction by modulating NOD2 signaling via RIPK2. Furthermore, miR-146a expression within CD4$^+$ T cells has previously been shown to directly limit IL-17 production and Th17 differentiation in other contexts[56]. However, our results suggest that miR-146a could operate further upstream at the myeloid cell level, to limit the availability of IL-17-inducing cytokines, thereby limiting IL-17 not only from CD4$^+$ T cells but also from other major sources of IL-17 producers in the colon, such as γδ T cells and ILCs[15].

IL-17 has been shown to exert tumorigenic signaling directly within IECs to promote CRC development[7]. However, the regulatory factors that limit tumorigenic IL-17R signaling in IECs remain unclear. To study miR-146a within IECs, we have leveraged multiple in vivo models, which recapitulate key aspects of colon tumorigenesis in humans, including tumorigenic mutation exacerbated by chronic inflammation in the tumor microenvironment[27].

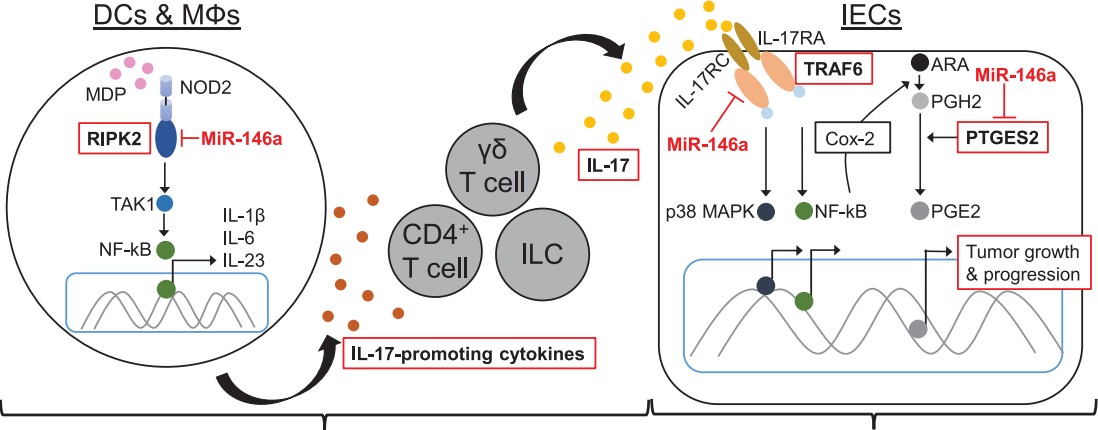

**Fig. 7 Molecular mechanisms involved in miR-146a-mediated prevention of tumor-promoting inflammation in CRC.** Our data demonstrate miR-146a prevents intestinal inflammation and CRC development by two interlinked mechanisms: (1) by limiting myeloid cell-mediated inflammatory IL-17 production; and (2) by inhibiting tumorigenic IL-17R signaling in IECs. Specifically, miR-146a targets RIPK2 in DCs/MΦs and limits NOD2 signaling, thereby limiting IL-17-promoting cytokine secretion and induction of IL-17 producers in the colon. In addition to limiting IL-17 levels, miR-146a interferes with tumorigenic IL-17R signaling within IECs by targeting TRAF6, thereby inhibiting p38 MAPK and NF-κB signaling. We also show that miR-146a in IECs may reduce PGE2 in the tumor microenvironment to further suppress CRC growth by targeting IL-17R-induced Cox-2 and by directly inhibiting PTGES2, an enzyme responsible for PGE2 synthesis.

Our findings establish a beneficial role for miR-146a expression in limiting responsiveness to tumorigenic IL-17. (Furthermore, our work provides additional support to studies showing a pathogenic role for IL-17a in DSS-induced colitis models[57,58]) Specifically, our results identify a critical role for miR-146a in limiting IL-17R-mediated tumorigenic signaling by targeting TRAF6, which we show for the first time to our knowledge in IECs in vivo. MiR-146a[−/−] IECs express higher levels of TRAF6 and respond robustly to IL-17 stimulation through activation of downstream signaling molecules NF-κB and p38 MAPK. Although treatment with anti-IL-17a rescues CRC severity in IEC-miR-146a[−/−] mice, it remains possible that other miR-146a-dependent mechanisms independent of IL-17R signaling may contribute to CRC resistance, as effective IL-17 neutralization has been shown to ameliorate CRC across multiple phenotypes[5,7,52]. Future experiments might leverage more nuanced, physiological approaches to partially downregulate heighted IL-17R signaling in these mice, such as IL-17R heterozygosity[7], to disentangle the possible role of additional miR-146a-dependent mechanisms.

Besides regulating NF-κB and p38 MAPK, our data suggest miR-146a also modulates PGE2 in tumor cells, a major driver in CRC development and progression[46]. The prominent role of PGE2 in CRC development and progression has been supported by numerous mouse model and patient studies[48,59]. Elevated levels of PGE2 and Cox-2, an upstream inducer of PGE2, have been found in a majority of adenocarcinomas and are associated with worse survival in CRC patients[46]. In fact, long-term use of the nonsteroidal anti-inflammatory drug (NSAID), aspirin, which primarily targets the PGE2 synthesis pathway by inhibiting Cox-2, is effective in reducing the risk of developing CRC[60]. In addition, selective inhibition of PGE2 via Cox-2 is beneficial in reducing polyp burden in a subset of CRC patients[61]. However, the enthusiasm for Cox-2 inhibitors has dampened due to a high incidence of cardiovascular toxicity[62]. Although PGE2 is heavily implicated in CRC pathogenesis and treatment, the physiologic regulatory mechanisms that control PGE2 levels within tumor cells to prevent their growth are not fully understood. Our data suggest miR-146a acts as a molecular brake on the PGE2 synthesis pathway to modulate tumorigenic PGE2 levels within IECs. Specifically, miR-146a limits PGE2 levels within

IECs through two distinct mechanisms: (1) by limiting IL-17R-TRAF6-induced Cox-2 levels and (2) by directly inhibiting PTGES2, an enzyme that converts PGH2 to PGE2[48], to confer CRC resistance. Accordingly, miR-146a[−/−] IECs and CRC tissues exhibit enhanced levels of PGE2 signaling. Consistent with enhanced IL-17R signaling, including NF-κB, p38 MAPK, and PGE2, immunostaining for Ki67 reveals increased proliferation of IECs in tumors from miR-146a[−/−] mice. Interestingly, some recent studies using CRC cell line systems suggest miR-146a may in fact limit the proliferation and metastatic potential of CRC[63,64]. These studies may further validate our orthotopic in vivo observations that deficiency or overexpression of miR-146a can confer CRC susceptibility or protection, along with higher or lower IEC proliferation, respectively.

CRC, both colitis-associated and sporadic cases, remains a leading cause of cancer-related death[65]. Patients are often resistant to traditional immunotherapy and chemotherapy, highlighting the urgent need for better options[10,11]. IL-17-neutralizing antibody has been shown to ameliorate CRC in preclinical models and even to abrogate chemotherapy resistance[7,10]. Our approach using miR-146a mimic to ameliorate colonic inflammation and CRC may offer an attractive alternative, as it limits both IL-17 induction via action within myeloid cells, and limits IL-17 responsiveness via action within IECs. Numerous studies linking miR-146a expression and polymorphisms to clinical outcomes in CRC lend high translational value to miR-146a as a therapeutic target[23,25,66,67]. This is further supported by our finding that mice deficient in miR-146a develop severe CRC, and that overexpression of miR-146a using miR-146a mimic ameliorates CRC severity. Recent advances in preclinical and clinical development of miRNA mimic-based treatments and targeted delivery systems support the feasibility of such a miRNA mimic-based therapy[68,69]. In this report, we have also shown therapeutic benefit using a combination of small molecule drugs to inhibit myeloid and IEC-miR-146a targets, RIPK2 and TRAF6, which represents another exciting more traditional therapeutic approach[70]. Further work is needed to evaluate these miR-146a-related therapeutic strategies and delivery methods, including their use as adjuvants to chemotherapy. In conclusion, miR-146a is of unique therapeutic significance, because it constitutes a

single powerful target in CRC that appears to modulate multiple pathways converging on tumorigenic IL-17 signaling.

## Methods

**Mice.** C57BL/6 WT, Apc[Min], miR-146a[−/−], miR-146a[flox/flox], LysM[cre], Villin[cre], and IL-17[GFP] mice were purchased from Jackson Laboratory. Myeloid-miR-146a[−/−] mice were obtained by crossing miR-146a[flox/flox] with LysM[cre] mice. IEC-miR-146a[−/−] mice were created by crossing miR-146a[flox/flox] with Villin[cre] mice. All mice were age- (6–10 weeks old at the start of experiments) and sex-matched. Littermate controls were used where appropriate. Both male and female mice were used in this study. Mice were maintained in specific pathogen-free animal facilities at the Harvard Institutes of Medicine at Harvard Medical School and the Hale Building for Transformative Medicine at Brigham and Women's Hospital (Boston, MA). Mice were maintained at 20–25 °C, 50–70% humidity, and a 12 h light cycle with light phase beginning at 7 a.m. and ending at 7 p.m. Mice were housed with food and water ad libitum. All experiments were in accordance with guidelines from the Institutional Animal Care and Use Committee at Brigham and Women's Hospital.

**DSS-induced colitis.** Induction of colitis was performed using a previously described protocol[71]. In brief, mice received 2–3% (36,000–50,000 M.Wt.) DSS (MP Biomedicals) in drinking water for 7 days. Body weight loss as a clinical sign of colitis was recorded daily. Mice that showed excessive body weight loss (>25%) or signs of rectal prolapse were euthanized. Final body weight scores at the time of euthanasia were recorded for the remainder of monitoring where appropriate. For histopathological analyses, colons were excised, flushed with phosphate-buffered saline (PBS), opened longitudinally, arranged in Swiss rolls, and fixed in 10% buffered formalin. They were then embedded in paraffin, sectioned (6–10 mm), and stained with hematoxylin/eosin according to standardized protocols. Sections were blindly scored by a pathologist at the Harvard Rodent Histopathology Core Facility from 0 to 4 based on degree of ulceration, loss of crypt architecture, and inflammatory infiltrates. Slides were imaged using an Aperio eSlide Scanner (Leica Biosystems).

**FITC-dextran assay.** On day 7 after DSS administration, colitic mice were deprived of food and water for 6 h, then given FITC-dextran (Sigma-Aldrich; 400 mg/kg b.w) by oral gavage. Four hours later, mice were euthanized and immediately bled via cardiac puncture to collect the sera. Fluorescence intensity of the sera was measured by the GloMax Explorer Multimode Microplate Reader (Promega).

**AOM/DSS-induced CRC.** Induction of CRC was performed using a previously described protocol[27]. Briefly, mice were injected intraperitoneally (i.p.) with AOM at a concentration of 10 mg/kg body weight. One week after AOM injection, mice were treated with 2% DSS in drinking water for 7 consecutive days, which was followed by 14 days of regular drinking water. This DSS treatment was repeated for two additional cycles. After at least 10 weeks, the colon was removed, flushed with PBS, opened longitudinally, and macroscopic tumors were visualized with Alcian blue stain and counted.

**BM chimeras.** Recipient CD45.1 WT and CD45.2 miR-146a[−/−] mice were irradiated (2 × 600 rad) and injected intravenously with WT or miR-146a[−/−] 5 × 10[6] BM cells. Six weeks after BM transplantation, the reconstitution of BM cells was confirmed by flow cytometry. The confirmed chimeric recipient mice were given DSS in drinking water as described.

**Treatment of colitis and CRC mice.** For anti-IL-17 treatment, AOM/DSS-treated mice were injected i.p. with isotype control (clone MOPC-21, Bio X Cell, Cat#BE0083) or anti-IL-17 antibody (clone 17F3, Bio X Cell, Cat#BE0173) (500 μg/mouse) twice a week for the entire duration of the CRC until euthanasia. For miR-146a mimic treatment, AOM/DSS-treated mice were injected i.p. with control or miR-146a mimic (5 mg/kg body weight) in RNA in vivo transfection reagent (Biotool) twice a week for the entire duration of the CRC until euthanasia. For RIPK2 and TRAF6 inhibitor treatment, AOM/DSS-treated mice were injected i.p. with (5 mg/kg body weight) of RIPK2 or TRAF6 inhibitors (EMD Millipore) alone or in combination twice a week for the entire duration of CRC until euthanasia. dimethylsulfoxide diluted in PBS was used as a vehicle control for RIPK2 and TRAF6 inhibitors.

**Generation and isolation of DCs, MΦs, T cells, and IECs.** DCs and MΦs were derived from BM progenitor cells as previously described[72]. For DCs, femoral and tibial cells were collected in DC culture medium (Iscove's modified Dulbecco's medium (IMDM), 10% heat-inactivated fetal bovine serum (FBS), 100 U/mL penicillin, 100 μg/mL streptomycin, 50 μM 2-mercaptoethanol, 20 ng/mL granulocyte-macrophage colony-stimulating factor, and 10 ng/mL IL-4) and seeded in 24-well plates at a density of 1 × 10[6] cells/ml/well. Culture medium was replaced with fresh medium after 3 days. On days 5–6, dislodged cells were used as

BM-derived DCs. For MΦs, BM progenitor cells were cultured in MΦ culture medium (IMDM medium, 10% heat-inactivated FBS, 100 U/mL penicillin, 100 μg/mL streptomycin, 50 μM 2-mercaptoethanol, 20 ng/mL macrophage colony-stimulating factor). Culture medium was replaced with fresh medium after 3 days. On days 5–6, adherent cells were used as BM-derived MΦs.

Immune cells and IECs from the colonic LP of CRC mice were isolated from colonic segments as previously described[73]. In brief, colons were collected, opened longitudinally, and washed to eliminate fecal contents. To remove epithelial cells, colons were then incubated at 37° in a 1 mM of dithiothreitol toxin cocktail, while shaking at 200 r.p.m. for 20 min. IECs were then fluorescence-activated cell sorting (FACS)-sorted from the supernatant of this fraction[45]. To isolate LP immune cells, colons were then washed again, chopped into small pieces, and incubated at 37° in a cocktail containing liberase (Thermolysin Low) TL and DNAase, while shaking at 120 r.p.m. for 45 min. To obtain a single-cell suspension, digested colons were then passed through a series of 100 and 40 μm nylon mesh filters with the flat end of a 1 mL syringe. T cells and myeloid cells were then FACS-sorted from this fraction[35].

**Coculture assays.** BM-derived DCs were prestimulated with MDP (10 μg/mL) for 20 h, then washed before coculture. For DC-CD4[+] T-cell cocultures, murine spleens and lymph nodes were ground with the flat end of a 1 mL syringe and passed through a 70 μm nylon mesh filter. Splenocytes were washed in PBS and erythrocytes were lysed by incubation in Ammonium-Chloride-Potassium lysing buffer for 5 min. Naive CD4[+] T cells were isolated from the spleen and lymph node (LN) of IL-17[GFP] mice via magnetic selection according to the manufacturer's instructions (Naive CD4[+] T Cell Isolation Kit, Miltenyi Biotec). For DC-γδ T-cell cocultures, γδ T cells were first enriched from the spleens and LNs of IL-17[GFP] mice via magnetic selection according to the manufacturer's instructions (γδ[+] T Cell Isolation Kit, Miltenyi Biotec), then FACS-sorted (7AAD[−]CD3[+]γδ[+]) to obtain a pure population. Naive CD4[+] T cells or γδ T cells were subsequently cocultured with prestimulated CD11c[+] BMDCs from WT and miR-146a[−/−] mice at a 1 : 1 DC : CD4[+] T or DC : γδ T-cell ratio for 5 days with soluble anti-CD3 (clone 17A2, Biolegend, Cat#100202) and anti-CD28 (clone 37.51, Biolegend, cat#102102) (1 μg/mL), and low-dose Th17-polarizing conditions in the presence of transforming growth factor-β (0.5 ng/mL), IL-6 (10 ng/mL), and anti-IFN-γ (clone XMG1.2, Biolegend, Cat#505834) (10 μg/ml) in U-bottom 96-well plates. For DC-ILC cocultures, murine mesenteric LNs (MLNs) were ground with the flat end of a 1 mL syringe and passed through a 70 μm nylon mesh filter. Total ILCs were FACS-sorted (7AAD[−]Lin[−]CD127[+]) from the MLNs of IL-17[GFP] mice and cocultured with CD11c[+] BM-derived DCs from WT and miR-146a[−/−] mice at a 1 : 1 DC : ILC cell ratio for 5 days under the low-dose Th17-polarizing conditions as described above.

**Enzyme-linked immunosorbent assay.** Cell-free supernatants were assayed for PGE2 and IL-17 by enzyme-linked immunosorbent assay (ELISA) using mouse PGE2 (Abcam) and IL-17 ELISA kits (ThermoFisher), respectively. For PGE2 ELISAs, miR-146a-sufficient (WT) and miR-146a-deficient (−/−) IEC lines, (CMT-93), were activated with IL-17a (50 ng/mL) for 48 h. Cell-free supernatants were then assayed for PGE2 according to the manufacturer's instructions and the absorbance at 405 nm was measured by the GloMax Explorer Multimode Microplate Reader (Promega). For IL-17 ELISAs, cell-free supernatants were collected from DC-γδ T cell and DC-ILC cocultures on day 5, and assayed for IL-17 according to the manufacturer's instructions. The absorbance at 450 nm was measured by the GloMax Explorer Multimode Microplate Reader (Promega).

**Cell lines.** RAW 264.7 and CMT-93 cells were obtained from the American Type Culture Collection. For the generation of stable IEC lines, CMT-93 cells were transfected with control or miR-146a-silencing lentiviral vectors (Applied Biological materials, Inc). Transfected GFP[+] cells were FACS-sorted and then clonally expanded via single-cell cloning.

**Luciferase assays.** To generate luciferase reporters for RIPK2, TRAF6, and PTGES2, fragments of 3′-UTR, which contained corresponding putative miR-146a-binding site, were cloned into the NotI and XhoI sites of psiCHECK-2, downstream of *Renilla* luciferase. RAW 264.7 cells were used for the RIPK2 reporter and CMT-93 cells were used for the TRAF6 and PTGES2 reporters. RAW 264.7 and CMT-93 cells were grown in Dulbecco's modified Eagle's medium supplemented with 10% FBS and were co-transfected with psiCHECK-2 vector containing 3′-UTR variants and either control or miR-146a mimic oligos. One or two days later, luciferase activities were measured with Dual-Glo Luciferase Assay System (Promega) and *Renilla* luciferase activity was normalized to *Firefly* luciferase activity.

**Immunoblotting.** CRC tissues and FACS-sorted IECs were homogenized in RIPA buffer (50 mM Tris-HCl pH 7.4, 150 mM NaCl, 1% NP-40), 1× protease inhibitor cocktail (Roche Applied Science), and 1× phosphatase inhibitor cocktail (Sigma-Aldrich). Equal amounts of protein (25 μg) were resolved by polyacrylamide gel electrophoresis. Proteins were transferred onto a nitrocellulose membrane and immunoblotting was performed with mouse monoclonal anti-RIPK2 (clone 6F7,

Sigma-Aldrich, Cat#SAB1404621, dilution 1 : 500); TRAF6 (clone D21G3, Cell Signaling Technology, Cat# 8028 S, dilution 1 : 500); phospho-p65 (clone 93H1, Cell Signaling Technology, Cat#3033, dilution 1 : 1000); p65 (clone D14E12, Cell Signaling Technology, Cat#8242, dilution 1 : 1000); pp38 (clone D3F9, Cell Signaling Technology, Cat#4511, dilution 1 : 1000); IKKα (polyclonal, Cell Signaling Technology, Cat#2682S, dilution 1 : 500); RelB (clone D7D7W, Cell Signaling Technology, Cat#10544, dilution 1 : 500); p38 MAPK (clone D13E1, Cell Signaling Technology, Cat#8690, dilution 1 : 1000); β-catenin (clone D10A8, Cell Signaling Technology, Cat#8480, dilution 1 : 1000); Cox-2 (clone D5H5, Cell Signaling Technology, Cat#12282, dilution 1 : 1000); PTGES2 (clone OTI2C3, OriGene, Cat#TA505412, dilution 1 : 250); c-Rel (clone 1RELAH5, ThermoFisher Scientific, Cat#14-6111-82, dilution 1 : 1000); glyceraldehyde 3-phosphate dehydrogenase (GAPDH) (clone GAPDH-71.1, Sigma-Aldrich, Cat#G8795, dilution 1 : 3000); β-actin (clone AC-40, Sigma-Aldrich, Cat#A4700, dilution 1 : 3000); and α-tubulin (clone B-5-1-2, Sigma-Aldrich, Cat#T6074, dilution 1 : 3000) and antibodies. Gels were run in parallel and probed for the protein of interest. Loading controls or basal proteins were probed after stripping wherever possible. Otherwise, gels were run in in parallel. The figures were made after compiling these data. Images were captured with the Bio-Rad ChemiDoc MP imaging system using Image Lab v1.

**miRNA-pulldown assays**. Biotin-based pulldown assays to validate miR-146a targets were performed using a previously described protocol[74]. In brief, CMT-93 and RAW 264.7 cells were transfected with 3′ biotin-labeled scrambled (ctrl) or miR-146a mimic (ThermoFisher Scientific) at a final concentration of 30 nM using a siPORT Neofx (Ambion) reverse-transfection method per the manufacturer's protocol and cultured for 72 h. Cells were then washed with PBS, lysed with buffer (20 mM Tris pH 7.5, 100 mM KCl, 5 mM MgCl$_2$, 0.3% NP-40, 50 Units RNase OUT, 1 : 100 complete protease inhibitor), and cytoplasmic extract was isolated. Ten percent of cytoplasmic extract was saved for input in 750 μL Trizol LS. The remaining cytoplasmic extract was added to anti-Biotin (clone D5A7, Cell Signaling Technology, Cat#5597) followed by streptavidin-coated magnetic beads to pull down target mRNA–miRNA bead complexes. Total RNA was isolated from the input and the mRNA–miRNA bound bead complexes, and miRNA and mRNA targets were quantified by synthesizing cDNA followed by qPCR. Pulldown abundance was normalized to the input RNA.

**Flow cytometry**. Surface staining for flow cytometry was performed according to the manufacturer's instructions. Doublets (identified via forward scatter (FSC-H) vs. FSC-W) and dead cells (identified via 7AAD or fixable Aqua Live/Dead Cell Stain Kit (ThermoFisher Scientific)) were excluded from analyses. Intracellular staining for Foxp3 was performed using the FoxP3/Transcription Factor Staining Buffer Set (eBioscience) according to the manufacturer's instructions. Intracellular staining for IL-17A was performed using the BD Cytofix/Cytoperm™ Fixation/ Permeabilization Kit according to the manufacturer's instructions. Intracellular staining for TRAF6, phospho-p38 MAPK, and phospho-p65 was performed using the BD Phosflow Lyse/Fix Buffer according to the manufacturer's instructions.

The following antibodies were used for flow cytometry: anti-CD3-FITC (clone 17A2, Biolegend, Cat#100204, dilution 1 : 100); anti-CD4-APC (clone GK1.5, Biolegend, Cat#100412, dilution 1 : 100); anti-CD4-PE (clone GK1.5 Biolegend, Cat#100408, dilution 1 : 100); anti-CD4-Brilliant Violet 421 (clone GK1.5, Biolegend, Cat#100443, dilution 1 : 100); anti-IL-17A-PE (clone TC11-18H10.1, Biolegend, Cat#506904, dilution 1 : 100); anti-IL-17A-APC (clone TC11-18H10.1, Biolegend, Cat#506916, dilution 1 : 100); anti-CD11c-APC (clone N418, Biolegend, Cat#117310, dilution 1 : 100), anti-CD11c-PE (clone N418, Biolegend, Cat#117308, dilution 1 : 100); anti-CD11b-APC (clone M1/70, Biolegend, Cat#101212, dilution 1 : 100); anti-F4/80-PE/Cyanine 7 (clone BM8, Biolegend, Cat#123114, dilution 1 : 100), anti-I-A/I-E-PE (clone M5/114.15.2, Biolegend, Cat#107608, dilution 1 : 100); anti-TCR γ/δ-APC (clone GL3, Biolegend, Cat#118116, dilution 1 : 100); anti-CD326(Ep-CAM)-APC (clone G8.8, Biolegend, Cat#118214, dilution 1 : 100); anti-CD326(Ep-CAM)-PerCP/Cyanine 5.5 (clone G8.8, Biolegend, Cat#118220, dilution 1 : 100); anti-CD326 (Ep-CAM)-PE (clone VU1D9, Cell Signaling, Cat#8995, dilution 1 : 100); anti-CD45-FITC (clone 30-F11, Biolegend, Cat#103108, dilution 1 : 100); anti-CD45-APC (clone 30-F11, Biolegend, Cat#103112, dilution 1 : 100); anti-CD45-PerCP (clone 30-F11, Biolegend, Cat#103130, dilution 1 : 100); anti-CD45.1-PE/Cyanine 7 (clone A20, Biolegend, Cat#110730, dilution 1 : 100); anti-CD45.2-APC (clone 104, Biolegend, Cat#109814, dilution 1 : 100); anti-Lineage Cocktail-Brilliant Violet 421 (clone 17A2, Biolegend, Cat#133311, dilution 1 : 100); anti-CD127-PE (clone A7R34, Biolegend, Cat#135010, dilution 1 : 100); anti-p38 MAPK-PE (clone 36/p38 (pT180/pY182), BD Biosciences, Cat#612565, dilution 1 : 6); anti-TRAF6-PE (clone 210412, Abcam, Cat#ab210412, dilution 1 : 100); anti-Phospho-NF-κB-p65 (clone 93H1, Cell Signaling, Cat#5733, dilution 1 : 100); and anti-CD31-PerCP/Cyanine 5.5 (clone 390, Biolegend, Cat#102420, dilution 1 : 100).

**RNA isolation, cDNA synthesis, and quantitative real-time PCR**. Total RNA was isolated from cell pellets using the RNeasy Micro or Mini Kit (Qiagen) according to the manufacturer's instructions and was stored at −80 °C. First-strand cDNA synthesis was performed for each RNA sample for up to 1 μg of total RNA using Taqman reverse-transcription reagents. cDNA was amplified via sequence-specific FAM-labeled primers as follows: IL-17a, Mm00439618_m1; IL-23,

Mm01160011_m1; IL-6, Mm99999064_m1; IL-1β, Mm01336189_m1; PTGES2, Mm00460181_m1; RIPK2, Mm00446815_m1; CSF1, Mm00432686_m1; CSF2, Mm01290062_m1; CCL2, Mm00441242_m1 (Applied Biosystems), and real-time PCR mix (Applied Biosystems) on an ABI7500 cycler. Gapdh, Mm99999915_g1, was used as an endogenous control to normalize for differences in the amount of total RNA in each sample. All quantitative real-time PCR reactions were performed in duplicate. All values were calculated as fold changes relative to the expression of GAPDH, then presented as fold changes from relevant experimental conditions.

**Data collection and analyses software**. For data collection, Leica Biosystems Aperio Digital Pathology Slide Scanner was used to acquire microscopy images. BD (Becton Dickinson) Biosciences FACSDiva v8.0.2 was used to acquire flow cytometry data. TargetScan 7.1 and RNAhybrid 2.2 online algorithms were used to acquire miRNA-146a and miRNA-146a mRNA target sequences for binding predictions. Applied Biosystems QuantStudio 7 was used to acquire real-time quantitative PCR data. Western blot images were captured with the Bio-Rad ChemiDoc MP imaging system using Image Lab v1. Fluorescence, absorbance, and luminescence were measured with the GloMax Explorer Multimode Microplate Reader running Glomax Explorer v3.1.0. For data analyses, BD (Becton Dickinson) Flowjo 10.7.1 was used to analyze flow cytometry data and prepare figures. Leica Biosystems Aperio Imagescope 12.4.3 was used to prepare microscopy images. Graphpad Prism 9.0.2 was used to generate figures and to perform statistical analyses. Adobe Illustrator v25.1 was used to arrange figure panels for publication. Microsoft Excel for Office 365 was used to prepare source data files.

**Statistical analyses**. Statistical analyses were performed using an unpaired two-tailed Student's $t$-test, or one- or two-way analysis of variance as indicated. A $p$-value $< 0.05$ was considered statistically significant. Data are generally presented as the mean $\pm$ SEM. Please see individual figure captions for more details.

**Reporting summary**. Further information on research design is available in the Nature Research Reporting Summary linked to this article.

## Data availability
The authors declare that all data are available in the article, Supplementary Information, or from the corresponding author upon reasonable request. Source data are provided with this paper.

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

## Acknowledgements

This study was supported by the National Institutes of Health (R01 AI435801 and R01 AI151953 to G.M.).

## Author contributions

L.G. and G.M. designed and performed experiments and analyzed data. A.A. performed western blotting experiments related to NOD2 and IL-17 signaling, analyzed miRNA-pulldown data for miR-146a targets, and generated stable CMT-93 IEC clones. M.F. performed FITC-dextran assays, DC-T cell/ILC cocultures, and ELISAs with assistance from R.R. G.G., B.K., and N.S. performed luciferase assays. C.K. was involved in the generation of bone marrow chimeric mice. R.K. and S.S. assisted with FACS sorting and qPCR of colonic lamina propria samples from CRC mice. L.G. and G.M. prepared the manuscript, with participation from M.F.

## Competing interests

The authors declare no competing interests.
