## [Peer Review File · Nature Communications]

Reviewers' comments:

Reviewer #1 (Remarks to the Author):

Reviewer comments for mir146 paper

The manuscript by Garo et al reported a novel role of miR146a in limiting tumor development in the gut. The authors employed powerful genetic tool to dissect the roles of miR146a in myeloid cells and IECs, and found that 1) miR146a protects mice from DSS-induced colitis by functioning in both myeloid cells and IECs; 2) miR146a inhibits the development of colitis-associated cancer by inducing cytokines that lead to increased IL-17 production; 3) miR146a also inhibits sporadic CRC development through a similar mechanism; 4) miR146a functions within gut epithelial cells to desensitize IL-17 signaling, thus preventing CRC development; and 5) treatment with miR146a analogues or TRAF6 and RIPK2 inhibitors significantly reduced intestinal tumor burden. Overall, the work was elegantly designed and carefully carried out, leading to important discovery of the anti-colitic and anti-cancer role of miR146a in the gut. The mechanistic study for miR146a was also well done.

This reviewer has one question with regards to the link between miR146a and IL-17 in colitis: It has been shown that IL-17 is protective during DSS-induced colitis. When lacking IL-17A or IL-17RA, mice are more susceptible to DSS-induced colitis compared to WT controls. The same is true for IL-6 and IL-23 knockout mice. In this manuscript, the authors show that miR146a inhibits IL-17 production and signaling. If this is a direct mechanism, we would expect less severe colitis upon loss of miR146a. Is it possible that miR146a also regulates other processes that impacts colitis, and that the upregulation of IL-17 and its inducing cytokines is a consequence of exacerbated colitis?

Minor suggestion:

Figure 2h, X axis. The authors may consider labeling "CD45+ Lin" instead of just "CD45" to better depict iLC. The figure legends are clear enough.

Reviewer #2 (Remarks to the Author):

The manuscript by Garo et al. examined the role of miR-146a in limiting IL-17-driven inflammation in the context of colorectal cancer (CRC) development. Previously, it has been shown that miR-146a controls IL-17 responses in a T cell-intrinsic manner (Bo et al., 2017; this ref should be cited). Here, the authors have further demonstrated that miR-146a can regulate IL-17 responses in the gut through limiting myeloid cell-derived IL-17-inducing cytokines via targeting RIPK2 and through restricting intestinal epithelial cell (IEC) responsiveness to colonic IL-17 via targeting TRAF6. In addition, the authors have also shown that miR-146a within IECs can further suppresses CRC by directly targeting PTGES2, an enzyme responsible for PGE2 synthesis. Overall, the manuscript is well-written and the results are generally of good quality. That said, there are several issues needed to be properly addressed in order to make a strong case to support their proposed model.

1. Based on the author's model, miR-146a in immune cells inhibits IL-17 production while miR-146a in IECs suppresses IL-17R signaling. To this end, in Fig. 1i-o, did the authors detect more IL-17 being produced in the intestinal tissue of WT (miR-146a^{-/-} BM) chimeric mice but not in miR-146a^{-/-} (WT BM) chimeric mice despite that both of them developed colitis?
2. It has been previously shown that LyzMcre does not promote efficient gene deletion in DCs (Abram et al., 2014). Therefore, it is uncertain as to why the authors chose to use LysMcre miR-146a fl/fl mice to study the impact of miR-146a deficiency on DCs (Fig. 2j). The fact that miR-146a might not be efficiently deleted in DCs from LyzMcre miR-146a fl/fl mice suggested that elevated productions of IL-17-promoting cytokines detected in DCs were likely due to the indirect

consequence of miR-146a deletion in other myeloid cell types (macrophages...?) rather than a direct effect of miR-146a ablation in DCs.

3. On the other hand, in most of the following-up studies (Fig. 2q-u), DCs from miR-146a^{-/-} rather than from LysMcre miR-146a fl/fl mice were examined. It is thus unclear as to how much did miR-146a deletion in DCs vs. non-DC myeloid cells contribute to the phenotypes observed in miR-146a^{-/-} mice. Could the same colitis/tumor phenotype still be observed when miR-146a ablation is restricted to DCs alone (i.e. using CD11cCre mice)?

4. The authors concluded that miR-146a limits IL-17-promoting cytokines in myeloid cells by targeting RIPK2. However, as NOD2 activates NFκB pathway and previously, several NFκB signaling molecules such as IKKα, c-Rel and RelB have also been shown to be regulated by miR-146 (Etzrodt et al., 2012; Cho et al., 2018; these two refs should also be cited). Are those miR-146a targets also dysregulated in miR-146a^{-/-} DCs? Can RIPK2 knockdown alone rescue the miR-146a^{-/-} DC phenotypes?

5. Likewise, as IL-17R signaling has already been shown to be crucial for CRC tumorigenesis, rather than using complete KO nor using anti-IL-17 Ab to completely block IL-17 signaling, can having IL-17R heterozygosity in IECs rescue Villin1cre miR-146a fl/fl mice disease phenotype?

RESPONSE TO REVIEWERS

Introduction:

We thank the *Nature Communications* editors and selected reviewers for their consideration of our manuscript and for their patience throughout this past challenging year.

Enclosed please find our revised manuscript, entitled “MicroRNA-146a Limits Tumorigenic Inflammation in Colorectal Cancer” (NCOMMS-20-04580). We thank the reviewers for their thorough and constructive feedback; the manuscript has clearly been strengthened in response to their comments and we believe that all issues raised by the reviewers have been addressed. The manuscript has also been significantly revised to align with *Nature Communications* guidelines. We hope that you will now find the manuscript acceptable for publication. Please note the following:

- To assist the reviewers, major changes/additions in the manuscript text that were added *in direct response to reviewer comments* have been highlighted in gray.
- To assist the reviewers, new data/figure panels that were added *in direct response to reviewer comments* have been boxed in a transparent yellow background. Please see below for an example, where panel **j** data were in the initial manuscript submission, while panel **k** data were added to this resubmission.

Below, we have provided a point-by-point response to the reviewers' comments along with a description of incorporated changes to the manuscript, including their locations. We opted not to embed new data directly into the reviewer responses due to the substantial number of new figure panels. (Please note that we have added an additional myeloid cell figure to accommodate these new data, thus shifting subsequent Main Figure numbers from the initial submission). Instead, we ask the reviewers to kindly view any new findings in context within the cited figures (which we have highlighted for their convenience).

Reviewer #1:

Introduction: The manuscript by Garo et al reported a novel role of miR146a in limiting tumor development in the gut. The authors employed powerful genetic tool to dissect the roles of miR146a in myeloid cells and IECs, and found that 1) miR146a protects mice from DSS-induced colitis by functioning in both myeloid cells and IECs; 2) miR146a inhibits the development of colitis-associated cancer by inducing cytokines that lead to increased IL-17 production; 3) miR146a also inhibits sporadic CRC development through a similar mechanism; 4) miR146a functions within gut epithelial cells to desensitize IL-17 signaling, thus preventing CRC development; and 5) treatment with miR146a analogues or TRAF6 and RIPK2 inhibitors significantly reduced intestinal tumor burden. Overall, the work was elegantly designed and carefully carried out, leading to important discovery of the anti-colitic and anti-cancer role of miR146a in the gut. The mechanistic study for miR146a was also well done.

We thank the reviewer for the encouraging feedback. Please see below for a point-by-point response to any questions/issues raised.

Comment 1a: This reviewer has one question with regards to the link between miR146a and IL-17 in colitis: It has been shown that IL-17 is protective during DSS-induced colitis. When lacking IL-17A or IL-17RA, mice are more susceptible to DSS-induced colitis compared to WT controls. The same is true for IL-6 and IL-23 knockout mice. In this manuscript, the authors show that miR146a inhibits IL-17 production

and signaling. If this is a direct mechanism, we would expect less severe colitis upon loss of miR146a. Is it possible that miR146a also regulates other processes that impacts colitis, and that the upregulation of IL-17 and its inducing cytokines is a consequence of exacerbated colitis?

We appreciate this interesting question and have divided our response into several parts to address each of the points raised above by the reviewer.

The reviewer is correct that some past work has pointed to a protective role for IL-17A in DSS-induced colitis in mice. Initial experiments using either anti-IL17-neutralizing antibodies¹, or IL-17A knockout mice², found more severe colitis. The reviewer then asks how these findings apply to our manuscript, where we show miR-146 limits IL-17A production and signaling, while protecting against DSS-induced colitis. Despite these previous reports on the protective effects of IL-17A, recent reports have pointed to a pathogenic role for IL-17A in DSS colitis. Several papers have associated IL-17A with various DSS colitis susceptibility phenotypes^{3,4}. Multiple groups have also directly tested IL-17A knockout mice and observed that they actually develop less severe DSS-induced colitis^{5,6} as well as less severe DSS-induced inflammation during CRC⁷. These observations appear to contradict the initial colitis study in IL-17A knockout mice which found they were protected². It is challenging beyond the scope of this reviewer response to reconcile these discrepancies, including other studies which found no effect of IL-17A⁸ on IL-17RA knockout⁹ on DSS-induced colitis. The differences may be partially due to varying intensities of DSS regimens, or even environmental factors involving animal facilities, as some IL-17 effects on colitis seem to be microbiota-dependent⁸. However, a broader look at the literature seems to indicate a pathogenic role for IL-17A-related signaling in general in DSS-induced colitis.

Consistent with a pathogenic role for IL-17A signaling in DSS-induced colitis, the IL-17A-inducing cytokines, IL-6 and IL-23, seem to promote disease. Neutralizing IL-6 attenuates colitis¹⁰, as does genetic deletion of IL-6¹¹. A study by Buonocore et al. shows IL-23 knockout and IL-23R knockout mice also develop milder DSS-induced colitis¹². The authors of this IL-23 study¹² addressed a previous contradictory report that IL-23 knockout mice were more susceptible to colitis¹³, stating that “Unfortunately, the data presented in that study did not reach statistical significance and were not followed up by any in-depth analysis, and therefore it is not possible to determine the mechanistic underpinnings of the divergent results”. Despite non-trivial remaining controversy¹⁴, IL-17A, IL-6, and IL-23 overall appear to play a pathogenic role in DSS-induced colitis.

In other non DSS-induced colitis mouse models, a large body of work also points to a pathogenic role for IL-17A^{6,15-20} (although not without exception²¹), and the IL-17A-inducing cytokines, IL-6 and IL-23²²⁻²⁶. This area has been reviewed extensively^{14,27}, and the overall consensus has been that the context-dependent pleiotropic effects of IL-17-related signaling in intestinal inflammation skew toward pathogenic, as reflected by therapeutic trials that have pursued IL-17A signal *inhibition* approaches in inflammatory bowel disease (IBD) patients²⁸. Although direct targeting of IL-17A via IL-17A blocking antibodies has not been effective in IBD clinical trials²⁸, blocking the IL-17-promoting cytokine, IL-23, has shown efficacy in patients²⁹. Elucidating IL-17A regulatory networks is of high clinical relevance and ongoing clinical work in this area is exploring previous setbacks to more effectively target these pathways³⁰.

Our paper examines how miR-146a signaling in the gut serves as a master regulator to inhibit both IL-17A production and IL-17A responsiveness. We primarily focus on colitis-associated colorectal cancer, in which the pathogenic role of highly expressed IL-17A^{31,7} is more unequivocal. IL-17A has been shown to promote a range of inflammation-driven CRC and inflammation-associated spontaneous CRC models³²⁻³⁷, including DSS-driven CRC^{7,34}.

In response to this interesting question posed by the reviewer, we now note in the **Discussion** section that our manuscript adds support to studies showing a pathogenic role for IL-17a in DSS^{3,4,5,6,7} and other^{6,15-20} colitis models, and cite key references^{5,6}.

Comment 1b: In this manuscript, the authors show that miR146a inhibits IL-17 production and signaling. If this is a direct mechanism, we would expect less severe colitis upon loss of miR146a.

Based on the references above which indicate a pathogenic role for IL-17A-related signaling in DSS-induced colitis, we would expect *more* severe colitis upon loss of miR-146a, which we show limits IL-17A signaling. This is what we observe in our data and show in our manuscript.

Comment 1c: Is it possible that miR146a also regulates other processes that impacts colitis and that the upregulation of IL-17 and its inducing cytokines is a consequence of exacerbated colitis?

As we and others have reviewed³⁸, miR-146a could regulate other inflammatory processes relevant to colitis, a possibility we acknowledge in the **Discussion** section of our manuscript. Profiling of colitis tissue in **Fig. 2A** shows enhanced IL-17A signaling along with gross elevation of other inflammatory markers. During colitis-associated CRC, the primary focus of our paper, we show that miR-146a control of IL-17A production and signaling is the predominant mechanism by which miR-146a protects against colitis, as neutralizing IL-17A abrogates CRC susceptibility in miR-146a-deficient mice (**Fig. 3s, t, 4x, y; Supplementary Fig. 4p, q**).

Furthermore, we show the upregulation of IL-17A and IL-17A-inducing cytokines in miR-146a-deficient mice is mediated by the direct effects of miR-146a (i.e. not solely an indirect consequence of exacerbated colitis). Specifically, we demonstrate this *in vitro* in **Fig. 3** by directly stimulating miR-146a-deficient myeloid cells to show they produce more IL-17-inducing cytokines, and can enhance IL-17 production from cocultured cell populations.

Comment 2: Minor suggestion: Figure 2h, X axis. The authors may consider labeling “CD45+ Lin” instead of just “CD45” to better depict iLC. The figure legends are clear enough.

We thank the reviewer for the suggestion. All representative FACS plots now showing ILCs (**Fig. 2h, Supplementary Fig. 2d**) are now labelled “(Lin⁻) CD45” on the x-axis for clarity.

Reviewer #2

Introduction: The manuscript by Garo et al. examined the role of miR-146a in limiting IL-17-driven inflammation in the context of colorectal cancer (CRC) development. Previously, it has been shown that miR-146a controls IL-17 responses in a T cell-intrinsic manner (Bo et al., 2017; this ref should be cited). Here, the authors have further demonstrated that miR-146a can regulate IL-17 responses in the gut through limiting myeloid cell-derived IL-17-inducing cytokines via targeting RIPK2 and through restricting intestinal epithelial cell (IEC) responsiveness to colonic IL-17 via targeting TRAF6. In addition, the authors have also shown that miR-146a within IECs can further suppresses CRC by directly targeting PTGES2, an enzyme responsible for PGE2 synthesis. Overall, the manuscript is well-written and the results are generally of good quality. That said, there are several issues needed to be properly addressed in order to make a strong case to support their proposed model.

We again thank the reviewer for the thoughtful and positive feedback. We have now cited in the **Discussion** this important reference by Bo Li et al. showing that miR-146a expression within CD4⁺ T cells directly limits IL-17 production and Th17 differentiation in other contexts³⁹. Our results suggest that miR-146a could operate further upstream at the myeloid cell-level to limit the availability of IL-17-inducing cytokines, thereby limiting IL-17 not only from CD4⁺ T cells, but also from other major sources of IL-17 producers in the colon, such as $\gamma\delta$ T cells and ILCs⁴⁰.

Please see below for a point-by-point response to any issues/questions raised.

Comment 1: Based on the author's model, miR-146a in immune cells inhibits IL-17 production while miR-146a in IECs suppresses IL-17R signaling. To this end, in Fig. 11-o, did the authors detect more IL-

17 being produced in the intestinal tissue of WT (miR-146a^{-/-} BM) chimeric mice but not in miR-146a^{-/-} (WT BM) chimeric mice despite that both of them developed colitis?

The reviewer asks an interesting question: Does miR-146a deletion within myeloid cells [WT (miR-146a^{-/-} BM) chimeric mice] enhance IL-17 levels while miR-146a deletion within IECs [miR-146a^{-/-} (WT BM) chimeric mice] does not?

In **Fig. 1** of our paper, we displayed bone marrow chimera colitis data to implicate general cellular compartments critical to miR-146a control of intestinal inflammation. To probe specific miR-146a mechanisms in inflammation-associated CRC, including regulation of IL-17, we then leveraged more robust/specific conditional knockout mice. To address your question within the context of our paper, we now show no difference in IL-17 production in CRC tissue or colonic lamina propria immune cells from IEC-miR-146a^{-/-} mice lacking miR-146a within intestinal epithelial cells (**Supplementary Fig. 4d, e**). Including these data has strengthened our transition to miR-146a mechanisms within IECs explored in **Fig. 4**: Enhanced IL-17R signaling and severe CRC observed in IEC-miR-146a^{-/-} mice is due to defective miR-146a signaling and promotion of IL-17 responsiveness within IECs, as these mice present with normal overall levels of IL-17. This contrasts with myeloid-miR-146a^{-/-} mice lacking miR-146a within myeloid cells (but intact within IECs), which present with elevated IL-17 levels (**Fig. 2f-i**) and severe CRC.

For reference, new data added to our resubmitted manuscript in response to this reviewer comment include **Supplementary Fig. 4d, e**.

Comment 2: It has been previously shown that LysMcre does not promote efficient gene deletion in DCs (Abram et al., 2014). Therefore, it is uncertain as to why the authors chose to use LysMcre miR-146a fl/fl mice to study the impact of miR-146a deficiency on DCs (Fig. 2j). The fact that miR-146a might not be efficiently deleted in DCs from LysMcre miR-146a fl/fl mice suggested that elevated productions of IL-17-promoting cytokines detected in DCs were likely due to the indirect consequence of miR-146a deletion in other myeloid cell types (macrophages...?) rather than a direct effect of miR-146a ablation in DCs.

The reviewer raises valid points regarding the study of dendritic cells (DCs) *in vivo* during CRC in our manuscript. Please allow us to clarify: we hypothesize that miR-146a-deficiency increases IL-17-promoting cytokines from myeloid cells, including *both* DCs and macrophages (MΦs), which then enhances IL-17 production and CRC development. Our proposed role for miR-146a within both populations was illustrated in our final summary model in **Fig 7**. Both MΦs and DCs play a critical role in CRC development, including the promotion of tumorigenic IL-17 in the colonic microenvironment^{41,42}.

After studying CRC susceptibility following global miR-146a deficiency (which deletes miR-146a within DCs, MΦs, and all other cell types), we then generated myeloid-miR-146a^{-/-} (LysM^{Cre}miR-146a^{fl/fl}) mice to study the impact of miR-146a deficiency more precisely in the myeloid compartment. As the reviewer cited, the excellent paper by Abrams et al. used reporter fate-mapping tools to compare the efficiency and specificity of different myeloid-Cre deleting strains⁴³. The reviewer is correct that LysM^{Cre} preferentially deletes genes in MΦs vs DCs⁴⁴, and that gut inflammation studies leveraging LysM^{Cre} mice typically emphasize the role of macrophages⁴⁵⁻⁵⁰. Of note, however, Abrams et al. show large differences in deletion patterns within each myeloid-Cre strain across tissues, and are careful to emphasize that “obviously, deletion efficiency and specificity could differ for any individual Cre line in different disease models” and that “there are changes in these populations during inflammatory and autoimmune disease settings, which could lead to changes in Cre-mediated deletion efficiency and specificity compared to naïve conditions.”⁴³ For example, one report suggested 50% deletion efficiency in tissue resident CD11c⁺ DCs using LysM-Cre reporter mice to study the lung, much higher than originally reported⁵¹. Others have also cautioned about overemphasizing myeloid-specific promoter deletion efficacies/specificities in DCs vs MΦs^{52,53}. Unfortunately, no perfect myeloid-Cre deleting mouse strain exists that completely and specifically deletes floxed genes of interest within MΦs and DCs. More work is needed to develop improved myeloid gene deletion tools, and more studies with sophisticated reporter and fate-mapping

mouse strains are needed to assess the tools we currently possess across a broader range of tissue types, disease contexts, and time courses.

When profiling myeloid cells during CRC for our miR-146a^{-/-} myeloid cell studies, we initially focused on DCs since DCs are professional antigen-presenting cells and major RIPK2-dependent regulators of IL-17 production in the gut²⁰. Specifically, we showed upregulation of IL-17-promoting cytokines and RIPK2 in colonic lamina propria DCs from both myeloid-miR-146a^{-/-} (**Fig. 2j, 3b**) and global miR-146a^{-/-} (**Supplementary Fig. 2g, 3b**) mice with CRC. We now present new profiling data from myeloid-miR-146a^{-/-} mice with CRC which shows upregulation of IL-17-promoting cytokines (**Fig. 2k**) and RIPK2 (**Fig. 3c**) in MΦs as well. We also expanded our DC *in vitro* studies (**Fig. 3h-j, n**) to include MΦs (**Fig. 3k-m, o**). Both miR-146a^{-/-} DCs and MΦs generated from the bone marrow of miR-146a^{-/-} mice show enhanced NOD2-RIPK2 signaling and IL-17-promoting cytokines in response to MDP stimulation. These new MΦ data provide better support for our model (**Fig. 7**) that miR-146a deletion within both MΦs and DCs directly promotes RIPK2, IL-17-promoting cytokines, and CRC susceptibility. They also better align with our LysM-Cre-deletion mediated approach which would affect both DCs and MΦs.

For reference, new data added to our resubmitted manuscript in response to this reviewer comment include **Fig. 2k, 3c, k-m, o; Supplementary Fig. 3d, f**.

Please see the following related **Reviewer 2 Comment 3** for further discussion on the direct effect of miR-146a ablation within DCs in myeloid-miR-146a^{-/-} mice.

Comment 3: On the other hand, in most of the following-up studies (Fig. 2q-u), DCs from miR-146a^{-/-} rather than from LysMcre miR-146a fl/fl mice were examined. It is thus unclear as to how much did miR-146a deletion in DCs vs. non-DC myeloid cells contribute to the phenotypes observed in miR-146a^{-/-} mice. Could the same colitis/tumor phenotype still be observed when miR-146a ablation is restricted to DCs alone (i.e. using CD11cCre mice)?

(Please begin with our response to the related **Reviewer 2 Comment 2** before proceeding).

The reviewer indicates an important gap. We observed increased IL-17-promoting cytokines in miR-146a-lacking DCs from both global miR-146a^{-/-} mice (**Supplementary Fig. 2g**) and myeloid-miR-146a^{-/-} (LysM^{Cre}miR-146a^{fl/fl}) mice (**Fig. 2j**) during CRC, suggesting a direct role for miR-146a ablation within DCs. However, in our original submission, our *in vitro* studies showing miR-146a deletion directly enhances IL-17-promoting cytokines had leveraged only DCs from miR-146a^{-/-} mice with a complete miR-146a deletion (**Fig. 3h-n**). Thus, we did not explicitly demonstrate that miR-146a ablation within DCs from myeloid-miR-146a^{-/-} mice was sufficient to directly enhance IL-17-promoting cytokines. In this manuscript resubmission, we now show that DCs and MΦs from myeloid-miR-146a^{-/-} mice present with enhanced RIPK2 and IL-17-promoting cytokines following MDP stimulation (**Supplementary Fig. 3c-f**), demonstrating a direct role for miR-146a ablation in both DCs and non-DC myeloid cells from this mouse strain as well.

These data are consistent with our hypothesis that miR-146a-deficiency increases IL-17-promoting cytokines from myeloid cells, including both DCs and MΦs, which enhances colonic IL-17 levels and CRC development (**Fig. 7**). Other deletions of miR-146a using myeloid-Cre strains reviewed in Abrams et al.⁴³ with differentially tailored MΦ/DC specificities, such as Cd11c-Cre to emphasize DCs (as suggested by the reviewer), should therefore lead to various extents of a similar phenotype: enhanced IL-17-promoting cytokines and CRC.

For reference, the data Reviewer 2 cites in the previous **Fig. 2q-u** can now be found in **Fig. 3h-j, n, r**. We have added an additional myeloid cell figure to accommodate new data, thus shifting subsequent Figure numbers from the initial submission. New data in our resubmitted manuscript in response to this reviewer comment include **Supplementary Fig. 3c-f**. Key references regarding LysM-Cre myeloid specificities have also been cited in the **Results** section for **Fig. 3**^{51,54}.

Comment 4: The authors concluded that miR-146a limits IL-17-promoting cytokines in myeloid cells by targeting RIPK2. However, as NOD2 activates NF- κ B pathway and previously, several NF- κ B signaling molecules such as IKK α , c-Rel and RelB have also been shown to be regulated by miR-146 (Etzrodt et al., 2012; Cho et al., 2018; these two refs should also be cited). Are those miR-146a targets also dysregulated in miR-146a^{-/-} DCs? Can RIPK2 knockdown alone rescue the miR-146a^{-/-} DC phenotypes?

This is an important question. Like other pattern-recognition receptors, NOD2 has also been shown to signal via the classical (canonical) NF- κ B pathway. NOD2-induced NF- κ B activation is mediated via RIPK2, an upstream canonical NF- κ B activator⁵⁵. In fact, RIPK2-deficient myeloid cells fail to activate NF- κ B in response to the NOD2 ligand, MDP^{55,56}. As discussed in our comments above, our original manuscript data demonstrated that RIPK2 was upregulated in miR-146a-deficient DCs (**Fig. 3b**). In this resubmission, our new data demonstrate that RIPK2 is also upregulated in miR-146a-deficient macrophages during CRC (**Fig. 3c**).

Additional new data in this resubmission also support that miR-146a targets RIPK2 to prevent MDP-NOD2-induced NF- κ B activation and IL-17-promoting cytokines in DCs and M Φ s: RIPK2 knockdown alone can abrogate MDP-induced inflammatory cytokines in miR-146a^{-/-} DCs and M Φ s (**Fig. 3p, q**). This is consistent with our *in vivo* finding that RIPK2 inhibitor given to miR-146a^{-/-} mice abrogates severe CRC development (**Fig. 6r, s**).

As suggested by the reviewer, we also examined the expression levels of other canonical (c-Rel) and non-canonical (RelB) NF- κ B subunits, as well as NF- κ B signaling molecules (IKK α), known to be regulated by miR-146a^{57,58} (**Supplementary Fig. 3g, h**). M Φ s from WT and miR-146a^{-/-} mice exhibited comparable levels of c-Rel and RelB, with the exception of IKK α being increased in miR-146a^{-/-} M Φ s. Although the increase in IKK α in miR-146a^{-/-} M Φ s could lead to enhanced NF- κ B signaling upon activation, the activation of IKK α is dependent on the upstream regulator, RIPK2, in the context of MDP-NOD2 signaling^{59,60}. This is consistent with our *in vitro* RIPK2 knockdown rescue experiments in MDP-stimulated miR-146a^{-/-} M Φ s described above (**Fig. 3p, q**).

In WT and miR-146a^{-/-} DCs, we did not observe a difference in the expression levels of IKK α and c-Rel, but found increased levels of RelB. However, we focused on RIPK2 and the canonical NF- κ B pathway because of its relevance to NOD2 signaling and to IL-17-promoting cytokines in myeloid cells²⁰. Indeed, neutralization of IL-17 abrogates CRC severity caused by miR-146a deletion within myeloid cells (**Fig. 3s, t**), and RIPK2 inhibition also ameliorates CRC susceptibility in miR-146a^{-/-} mice (**Fig. 6r, s**), supporting this focus. Together, our data suggest that miR-146a targets RIPK2, an upstream regulator of the canonical NF- κ B pathway, to limit NOD2-induced NF- κ B activation and IL-17-promoting cytokines in DCs and M Φ s.

For reference, key new data added to our resubmitted manuscript in response to this reviewer comment include **Fig. 3p, q; Supplementary Fig. 3g, h**. We have also now cited these important references^{57,58} and discussed these key points in the **Results** section for **Fig. 3**.

Comment 5: Likewise, as IL-17R signaling has already been shown to be crucial for CRC tumorigenesis, rather than using complete KO nor using anti-IL-17 Ab to completely block IL-17 signaling, can having IL-17R heterozygosity in IECs rescue Villin1cre miR-146a fl/fl mice disease phenotype?

The reviewer asks an important question about IL-17R signaling specifically within IECs mediating CRC susceptibility in IEC-miR-146a^{-/-} mice. For this manuscript, we used/generated a number of mouse strains, including miR-146a^{-/-}, myeloid-miR-146a^{-/-} (Lysm^{Cre}miR-146a^{fl/fl}) and IEC-miR-146a^{-/-} (Villin^{Cre}miR-146a^{fl/fl}) mice. Anti-IL-17 antibody allowed us to probe the pathogenic role of IL-17 uniformly across all three strains (**Fig. 3s, 3t, 4s, 4t; Supplementary Fig. 4p, 4q**). In our Villin^{Cre}miR-146a^{fl/fl} mice, multiple cell types express IL-17R⁶¹. However, it can be inferred that the rescue of CRC susceptibility in these mice following anti-IL-17 is due to miR-146a control of IL-17R signaling within IECs because of **1)** the conditional nature of the miR-146a deletion in IECs; **2)** new data (in response to **Reviewer 2**

Comment 1) showing that IEC-miR-146a^{-/-} mice do not express elevated IL-17, suggesting miR-146-mediated altered IL-17R sensitivity mediates CRC susceptibility (**Supplementary Fig. 3d, e**); and **3)** the excellent work by Wang et al. referenced by the reviewer showing that IL-17R deletion within IECs (not hematopoietic cells) ablates CRC³⁶. The value of our manuscript beyond the work by Wang et al. lies in revealing the IEC-intrinsic role of miR-146a in this process.

To illustrate the role of miR-146a in IL-17R signaling specifically within IECs more clearly, we have now added a large amount of new IEC-related miR-146a data. In our initial submission, we mostly studied IL-17R signaling using CRC tissue from global miR-146a^{-/-} mice. For this resubmission, we now profile CRC tissue and sorted IECs from IEC-miR-146a^{-/-} mice in the main **Fig. 4**, supplemented by CRC tissue and sorted IEC data from global miR-146a^{-/-} mice in **Supplementary Fig. 4**. All data support enhanced IL-17R signaling specifically within miR-146a^{-/-} deficient IECs, such that CRC severity is ameliorated upon treatment with anti-IL-17 (**Fig. 4x, y**).

We think these data from the mouse strains mentioned above, data from other mouse strains in the paper including Apc^{min/+}miR-146a^{-/-} mice and miR-146a bone marrow chimeras, as well as data from multiple treatment studies in **Fig. 6** (e.g. miR-146a mimic, RIPK2 inhibitor, TRAF6 inhibitor), altogether elucidate miR-146a mechanisms in IECs (and myeloid cells) without the need for generating an additional strain with a conditional deletion of IL-17R in miR-146a^{-/-} IECs.

For reference, new data added to our resubmitted manuscript in response to this reviewer comment include **Fig. 4d, l, k, l, o-r**; **Supplementary Fig. 4d-f, h, k, m-o**.

References

- 1 Ogawa, A., Andoh, A., Araki, Y., Bamba, T. & Fujiyama, Y. Neutralization of interleukin-17 aggravates dextran sulfate sodium-induced colitis in mice. *Clin Immunol* **110**, 55-62, doi:10.1016/j.clim.2003.09.013 (2004).
- 2 Yang, X. O. *et al.* Regulation of inflammatory responses by IL-17F. *J Exp Med* **205**, 1063-1075, doi:10.1084/jem.20071978 (2008).
- 3 Fina, D. *et al.* Regulation of gut inflammation and th17 cell response by interleukin-21. *Gastroenterology* **134**, 1038-1048, doi:10.1053/j.gastro.2008.01.041 (2008).
- 4 Medina-Contreras, O. *et al.* CX3CR1 regulates intestinal macrophage homeostasis, bacterial translocation, and colitogenic Th17 responses in mice. *J Clin Invest* **121**, 4787-4795, doi:10.1172/JCI59150 (2011).
- 5 Ito, R. *et al.* Involvement of IL-17A in the pathogenesis of DSS-induced colitis in mice. *Biochem Biophys Res Commun* **377**, 12-16, doi:10.1016/j.bbrc.2008.09.019 (2008).
- 6 Park, C. H., Lee, A. R., Ahn, S. B., Eun, C. S. & Han, D. S. Role of innate lymphoid cells in chronic colitis during anti-IL-17A therapy. *Sci Rep* **10**, 297, doi:10.1038/s41598-019-57233-w (2020).
- 7 Hyun, Y. S. *et al.* Role of IL-17A in the development of colitis-associated cancer. *Carcinogenesis* **33**, 931-936, doi:10.1093/carcin/bgs106 (2012).
- 8 Tang, C. *et al.* Suppression of IL-17F, but not of IL-17A, provides protection against colitis by inducing Treg cells through modification of the intestinal microbiota. *Nat Immunol* **19**, 755-765, doi:10.1038/s41590-018-0134-y (2018).
- 9 Maxwell, J. R. *et al.* Differential Roles for Interleukin-23 and Interleukin-17 in Intestinal Immunoregulation. *Immunity* **43**, 739-750, doi:10.1016/j.immuni.2015.08.019 (2015).
- 10 Sommer, J. *et al.* Interleukin-6, but not the interleukin-6 receptor plays a role in recovery from dextran sodium sulfate-induced colitis. *Int J Mol Med* **34**, 651-660, doi:10.3892/ijmm.2014.1825 (2014).
- 11 Naito, Y. *et al.* Reduced intestinal inflammation induced by dextran sodium sulfate in interleukin-6-deficient mice. *Int J Mol Med* **14**, 191-196 (2004).
- 12 Cox, J. H. *et al.* Opposing consequences of IL-23 signaling mediated by innate and adaptive cells in chemically induced colitis in mice. *Mucosal Immunol* **5**, 99-109, doi:10.1038/mi.2011.54 (2012).
- 13 Becker, C. *et al.* Cutting edge: IL-23 cross-regulates IL-12 production in T cell-dependent experimental colitis. *J Immunol* **177**, 2760-2764, doi:10.4049/jimmunol.177.5.2760 (2006).
- 14 Mizoguchi, A. Animal models of inflammatory bowel disease. *Prog Mol Biol Transl Sci* **105**, 263-320, doi:10.1016/B978-0-12-394596-9.00009-3 (2012).
- 15 Leppkes, M. *et al.* RORgamma-expressing Th17 cells induce murine chronic intestinal inflammation via redundant effects of IL-17A and IL-17F. *Gastroenterology* **136**, 257-267, doi:10.1053/j.gastro.2008.10.018 (2009).
- 16 Zhang, Z., Zheng, M., Bindas, J., Schwarzenberger, P. & Kolls, J. K. Critical role of IL-17 receptor signaling in acute TNBS-induced colitis. *Inflamm Bowel Dis* **12**, 382-388, doi:10.1097/O1.MIB.0000218764.06959.91 (2006).
- 17 Feng, T. *et al.* Th17 cells induce colitis and promote Th1 cell responses through IL-17 induction of innate IL-12 and IL-23 production. *J Immunol* **186**, 6313-6318, doi:10.4049/jimmunol.1001454 (2011).
- 18 Do, J. S., Vissers, A., Dong, C., Baldwin, W. M., 3rd & Min, B. Cutting edge: Generation of colitogenic Th17 CD4 T cells is enhanced by IL-17+ gammadelta T cells. *J Immunol* **186**, 4546-4550, doi:10.4049/jimmunol.1004021 (2011).
- 19 Wedebye Schmidt, E. G. *et al.* TH17 cell induction and effects of IL-17A and IL-17F blockade in experimental colitis. *Inflamm Bowel Dis* **19**, 1567-1576, doi:10.1097/MIB.0b013e318286fa1c (2013).
- 20 Ermann, J., Staton, T., Glickman, J. N., de Waal Malefyt, R. & Glimcher, L. H. Nod/Ripk2 signaling in dendritic cells activates IL-17A-secreting innate lymphoid cells and drives colitis in T-bet-/-

- .Rag2^{-/-} (TRUC) mice. *Proc Natl Acad Sci U S A* **111**, E2559-2566, doi:10.1073/pnas.1408540111 (2014).
- 21 O'Connor, W., Jr. *et al.* A protective function for interleukin 17A in T cell-mediated intestinal inflammation. *Nat Immunol* **10**, 603-609, doi:10.1038/ni.1736 (2009).
- 22 Yen, D. *et al.* IL-23 is essential for T cell-mediated colitis and promotes inflammation via IL-17 and IL-6. *J Clin Invest* **116**, 1310-1316, doi:10.1172/JCI21404 (2006).
- 23 Elson, C. O. *et al.* Monoclonal anti-interleukin 23 reverses active colitis in a T cell-mediated model in mice. *Gastroenterology* **132**, 2359-2370, doi:10.1053/j.gastro.2007.03.104 (2007).
- 24 Atreya, R. *et al.* Blockade of interleukin 6 trans signaling suppresses T-cell resistance against apoptosis in chronic intestinal inflammation: evidence in crohn disease and experimental colitis in vivo. *Nat Med* **6**, 583-588, doi:10.1038/75068 (2000).
- 25 Kullberg, M. C. *et al.* IL-23 plays a key role in Helicobacter hepaticus-induced T cell-dependent colitis. *J Exp Med* **203**, 2485-2494, doi:10.1084/jem.20061082 (2006).
- 26 Buonocore, S. *et al.* Innate lymphoid cells drive interleukin-23-dependent innate intestinal pathology. *Nature* **464**, 1371-1375, doi:10.1038/nature08949 (2010).
- 27 Conn, M. Animal model of IBD. *Animal Models for the Study of Human Disease* (2017).
- 28 Fitzpatrick, L. R. Inhibition of IL-17 as a pharmacological approach for IBD. *Int Rev Immunol* **32**, 544-555, doi:10.3109/08830185.2013.821118 (2013).
- 29 Kashani, A. & Schwartz, D. A. The Expanding Role of Anti-IL-12 and/or Anti-IL-23 Antibodies in the Treatment of Inflammatory Bowel Disease. *Gastroenterol Hepatol (N Y)* **15**, 255-265 (2019).
- 30 Verstockt, B., Van Assche, G., Vermeire, S. & Ferrante, M. Biological therapy targeting the IL-23/IL-17 axis in inflammatory bowel disease. *Expert Opin Biol Ther* **17**, 31-47, doi:10.1080/14712598.2017.1258399 (2017).
- 31 Shen W, L. W., Feigenbaum L, Hixon J, Durum S. Differential expression of IL-17A and IL-17F in inflammatory bowel disease/colon cancer using a new IL-17A/F-dual-color reporter mouse. *J Immunol* **186** (2011).
- 32 Chae, W. J. *et al.* Ablation of IL-17A abrogates progression of spontaneous intestinal tumorigenesis. *Proc Natl Acad Sci U S A* **107**, 5540-5544, doi:10.1073/pnas.0912675107 (2010).
- 33 Wu, S. *et al.* A human colonic commensal promotes colon tumorigenesis via activation of T helper type 17 T cell responses. *Nat Med* **15**, 1016-1022, doi:10.1038/nm.2015 (2009).
- 34 Kathania, M. *et al.* Itch inhibits IL-17-mediated colon inflammation and tumorigenesis by ROR-gammat ubiquitination. *Nat Immunol* **17**, 997-1004, doi:10.1038/ni.3488 (2016).
- 35 Grivennikov, S. I. *et al.* Adenoma-linked barrier defects and microbial products drive IL-23/IL-17-mediated tumour growth. *Nature* **491**, 254-258, doi:10.1038/nature11465 (2012).
- 36 Wang, K. *et al.* Interleukin-17 receptor signaling in transformed enterocytes promotes early colorectal tumorigenesis. *Immunity* **41**, 1052-1063, doi:10.1016/j.immuni.2014.11.009 (2014).
- 37 Housseau, F. *et al.* Redundant Innate and Adaptive Sources of IL17 Production Drive Colon Tumorigenesis. *Cancer Res* **76**, 2115-2124, doi:10.1158/0008-5472.CAN-15-0749 (2016).
- 38 Garo, L. P. & Murugaiyan, G. Contribution of MicroRNAs to autoimmune diseases. *Cell Mol Life Sci* **73**, 2041-2051, doi:10.1007/s00018-016-2167-4 (2016).
- 39 Li, B. *et al.* miR-146a modulates autoreactive Th17 cell differentiation and regulates organ-specific autoimmunity. *J Clin Invest* **127**, 3702-3716, doi:10.1172/JCI94012 (2017).
- 40 West, N. R., McCuaig, S., Franchini, F. & Powrie, F. Emerging cytokine networks in colorectal cancer. *Nat Rev Immunol* **15**, 615-629, doi:10.1038/nri3896 (2015).
- 41 Wang, K. & Karin, M. Tumor-Elicited Inflammation and Colorectal Cancer. *Adv Cancer Res* **128**, 173-196, doi:10.1016/bs.acr.2015.04.014 (2015).
- 42 Kather, J. N. & Halama, N. Harnessing the innate immune system and local immunological microenvironment to treat colorectal cancer. *Br J Cancer* **120**, 871-882, doi:10.1038/s41416-019-0441-6 (2019).

- 43 Abraham, C. & Medzhitov, R. Interactions between the host innate immune system and microbes in inflammatory bowel disease. *Gastroenterology* **140**, 1729-1737, doi:10.1053/j.gastro.2011.02.012 (2011).
- 44 Clausen, B. E., Burkhardt, C., Reith, W., Renkawitz, R. & Forster, I. Conditional gene targeting in macrophages and granulocytes using LysMcre mice. *Transgenic Res* **8**, 265-277, doi:10.1023/a:1008942828960 (1999).
- 45 Lin, N. *et al.* Myeloid Cell Hypoxia-Inducible Factors Promote Resolution of Inflammation in Experimental Colitis. *Front Immunol* **9**, 2565, doi:10.3389/fimmu.2018.02565 (2018).
- 46 Reindl, W., Weiss, S., Lehr, H. A. & Forster, I. Essential crosstalk between myeloid and lymphoid cells for development of chronic colitis in myeloid-specific signal transducer and activator of transcription 3-deficient mice. *Immunology* **120**, 19-27, doi:10.1111/j.1365-2567.2006.02473.x (2007).
- 47 Shimoda, M. *et al.* Epithelial Cell-Derived α 5 β 1 Integrin and Metalloproteinase-17 Confers Resistance to Colonic Inflammation Through EGFR Activation. *EBioMedicine* **5**, 114-124, doi:10.1016/j.ebiom.2016.02.007 (2016).
- 48 Ma, C. *et al.* Gasdermin D in macrophages restrains colitis by controlling cGAS-mediated inflammation. *Sci Adv* **6**, eaaz6717, doi:10.1126/sciadv.aaz6717 (2020).
- 49 Ahn, J., Son, S., Oliveira, S. C. & Barber, G. N. STING-Dependent Signaling Underlies IL-10 Controlled Inflammatory Colitis. *Cell Rep* **21**, 3873-3884, doi:10.1016/j.celrep.2017.11.101 (2017).
- 50 Zhang, H. *et al.* Myeloid ATG16L1 Facilitates Host-Bacteria Interactions in Maintaining Intestinal Homeostasis. *J Immunol* **198**, 2133-2146, doi:10.4049/jimmunol.1601293 (2017).
- 51 McCubbrey, A. L., Allison, K. C., Lee-Sherick, A. B., Jakubzick, C. V. & Janssen, W. J. Promoter Specificity and Efficacy in Conditional and Inducible Transgenic Targeting of Lung Macrophages. *Front Immunol* **8**, 1618, doi:10.3389/fimmu.2017.01618 (2017).
- 52 Hume, D. A. Applications of myeloid-specific promoters in transgenic mice support in vivo imaging and functional genomics but do not support the concept of distinct macrophage and dendritic cell lineages or roles in immunity. *J Leukoc Biol* **89**, 525-538, doi:10.1189/jlb.0810472 (2011).
- 53 Arnold-Schrauf, C., Berod, L. & Sparwasser, T. Dendritic cell specific targeting of MyD88 signalling pathways in vivo. *Eur J Immunol* **45**, 32-39, doi:10.1002/eji.201444747 (2015).
- 54 Abram, C. L., Roberge, G. L., Hu, Y. & Lowell, C. A. Comparative analysis of the efficiency and specificity of myeloid-Cre deleting strains using ROSA-EYFP reporter mice. *J Immunol Methods* **408**, 89-100, doi:10.1016/j.jim.2014.05.009 (2014).
- 55 Park, J. H. *et al.* RICK/RIP2 mediates innate immune responses induced through Nod1 and Nod2 but not TLRs. *J Immunol* **178**, 2380-2386, doi:10.4049/jimmunol.178.4.2380 (2007).
- 56 Goncharov, T. *et al.* Disruption of XIAP-RIP2 Association Blocks NOD2-Mediated Inflammatory Signaling. *Mol Cell* **69**, 551-565 e557, doi:10.1016/j.molcel.2018.01.016 (2018).
- 57 Etzrodt, M. *et al.* Regulation of monocyte functional heterogeneity by miR-146a and Relb. *Cell Rep* **1**, 317-324, doi:10.1016/j.celrep.2012.02.009 (2012).
- 58 Cho, S. *et al.* Differential cell-intrinsic regulations of germinal center B and T cells by miR-146a and miR-146b. *Nat Commun* **9**, 2757, doi:10.1038/s41467-018-05196-3 (2018).
- 59 Chen, F., Demers, L. M. & Shi, X. Upstream signal transduction of NF-kappaB activation. *Curr Drug Targets Inflamm Allergy* **1**, 137-149, doi:10.2174/1568010023344706 (2002).
- 60 Strober, W., Murray, P. J., Kitani, A. & Watanabe, T. Signalling pathways and molecular interactions of NOD1 and NOD2. *Nat Rev Immunol* **6**, 9-20, doi:10.1038/nri1747 (2006).
- 61 Murugaiyan, G. & Saha, B. Protumor vs antitumor functions of IL-17. *J Immunol* **183**, 4169-4175, doi:10.4049/jimmunol.0901017 (2009).

REVIEWERS' COMMENTS

Reviewer #1 (Remarks to the Author):

The manuscript by Garo et al reported a novel role of miR146a in limiting tumor development in the gut. The authors employed powerful genetic tool to dissect the roles of miR146a in myeloid cells and IECs, and found that 1) miR146a protects mice from DSS-induced colitis by functioning in both myeloid cells and IECs; 2) miR146a inhibits the development of colitis-associated cancer by inducing cytokines that lead to increased IL-17 production; 3) miR146a also inhibits sporadic CRC development through a similar mechanism; 4) miR146a functions within gut epithelial cells to desensitize IL-17 signaling, thus preventing CRC development; and 5) treatment with miR146a analogues or TRAF6 and RIPK2 inhibitors significantly reduced intestinal tumor burden. The revised work has addressed this Reviewer's concerns and is now in a much better shape for publication.

Reviewer #2 (Remarks to the Author):

The authors have committed quite an effort to address the concerns raised in my previous review. I do, however, remain not entirely convinced about the last point related to the actual contribution of miR-146a-dependent regulation of IL-17R signaling in IEC to the observed CRC phenotype. To this end, in their response, the authors provided three points arguing against the requirement of the experiment suggested in my previous review: 1) IEC-specific miR-146a deletion; 2) no increased IL-17 production in IEC-cKO mice; 3) a previous study suggesting the critical role of IL-17R signaling in IEC for CRC development. However, in my opinion, these three arguments did not alleviate but rather support the concern I originally had. As the authors also acknowledged, miR-146a could target many other inflammatory pathways relevant to colitis. Therefore, even within the IEC compartment, it is expected that in addition to heightened IL-17R responses, many other miR-146a-dependent processes could also be dysregulated. Moreover, as IL-17R signaling is essential for CRC development, it is also not surprising that neutralizing IL-17A would abrogate CRC susceptibility in mice with or without miR-146a in IECs. As such, the key question is whether or not "heightened" IL-17R signaling in IECs in the absence of miR-146a-mediated regulation is at least one of the major mechanisms contributing to the observed CRC phenotype. Therefore, to directly address this question, in my opinion, the authors should examine the potential effect of IL-17R heterozygosity in IECs on rescuing IEC-KO mice CRC phenotype instead of using anti-IL-17 Ab that would completely neutralize IL-17 signaling as stated in my original review.

RESPONSE TO REVIEWERS

Reviewer #1

The manuscript by Garo et al reported a novel role of miR146a in limiting tumor development in the gut. The authors employed powerful genetic tool to dissect the roles of miR146a in myeloid cells and IECs, and found that 1) miR146a protects mice from DSS-induced colitis by functioning in both myeloid cells and IECs; 2) miR146a inhibits the development of colitis-associated cancer by inducing cytokines that lead to increased IL-17 production; 3) miR146a also inhibits sporadic CRC development through a similar mechanism; 4) miR146a functions within gut epithelial cells to desensitize IL-17 signaling, thus preventing CRC development; and 5) treatment with miR146a analogues or TRAF6 and RIPK2 inhibitors significantly reduced intestinal tumor burden. The revised work has addressed this Reviewer's concerns and is now in a much better shape for publication.

Response to Reviewer#1

We again thank the reviewer for the assistance with our manuscript. It is much improved.

Reviewer #2 (Remarks to the Author):

The authors have committed quite an effort to address the concerns raised in my previous review. I do, however, remain not entirely convinced about the last point related to the actual contribution of miR-146a-dependent regulation of IL-17R signaling in IEC to the observed CRC phenotype. To this end, in their response, the authors provided three points arguing against the requirement of the experiment suggested in my previous review: 1) IEC-specific miR-146a deletion; 2) no increased IL-17 production in IEC-cKO mice; 3) a previous study suggesting the critical role of IL-17R signaling in IEC for CRC development. However, in my opinion, these three arguments did not alleviate but rather support the concern I originally had. As the authors also acknowledged, miR-146a could target many other inflammatory pathways relevant to colitis. Therefore, even within the IEC compartment, it is expected that in addition to heightened IL-17R responses, many other miR-146a-dependent processes could also be dysregulated. Moreover, as IL-17R signaling is essential for CRC development, it is also not surprising that neutralizing IL-17A would abrogate CRC susceptibility in mice with or without miR-146a in IECs. As such, the key question is whether or not "heightened" IL-17R signaling in IECs in the absence of miR-146a-mediated regulation is at least one of the major mechanisms contributing to the observed CRC phenotype. Therefore, to directly address this question, in my opinion, the authors should examine the potential effect of IL-17R heterozygosity in IECs on rescuing IEC-KO mice CRC phenotype instead of using anti-IL-17 Ab that would completely neutralize IL-17 signaling as stated in my original review.

Response to Reviewer#2

We again thank the reviewer for the thorough review and for commending the large scope of revisions we pursued to address issues raised by the reviewers. Our data showing heightened IL-17R signaling (in the absence of elevated IL-17 itself) in IEC-miR-146a^{-/-} with severe CRC, combined with the rescue of CRC severity in IEC-miR-146a^{-/-} mice with anti-IL-17, (along with other data, e.g. identification of miR-146a targets in the IL-17R pathway) suggest a strong role for miR-146a within IECs in controlling IL-17R signaling to mediate CRC resistance. However, the reviewer is certainly correct that other miR-146a processes may be playing a role, and that this may be revealed with more nuanced, physiological downregulation of heightened IL-17R signaling, such as by IL-17R heterozygosity. (Indeed, our manuscript demonstrates miR-146a direct targeting of PTGES2, an enzyme involved in PGE2 synthesis, as one such additional mechanisms). We now acknowledge this possibility in the discussion and emphasize it as a potential future direction. Please see the relevant text below copied for your reference.

Although treatment with anti-IL-17 rescues CRC severity in IEC-miR-146a^{-/-} mice, it remains possible that other miR-146a-dependent mechanisms independent of IL-17R signaling may contribute to CRC protection, as effective IL-17 neutralization has been shown to ameliorate CRC across multiple phenotypes¹⁻³. Future experiments might leverage more nuanced, physiological approaches to partially downregulate heightened IL-17R signaling in these mice, such as IL-17R heterozygosity¹, to disentangle the possible role of additional miR-146a-dependent mechanisms.

REFERENCES

1. Wang, K., *et al.* Interleukin-17 receptor a signaling in transformed enterocytes promotes early colorectal tumorigenesis. *Immunity* **41**, 1052-1063 (2014).
2. Chae, W.J., *et al.* Ablation of IL-17A abrogates progression of spontaneous intestinal tumorigenesis. *Proc Natl Acad Sci U S A* **107**, 5540-5544 (2010).
3. Housseau, F., *et al.* Redundant Innate and Adaptive Sources of IL17 Production Drive Colon Tumorigenesis. *Cancer Res* **76**, 2115-2124 (2016).